



# A 439-year daily discharge dataset (1861-2299) for the upper Yangtze River, China

Chao Gao[1], Buda Su[2], Valentina Krysanova[3], Qianyu Zha[1], Cai Chen[1], Gang Luo[1], Xiaofan Zeng[4], Jinlong Huang[5], Min Xiong[6], Liping Zhang[7], Tong Jiang[5]

[1]Department of Geography & Spatial Information Techniques, Ningbo University, Ningbo, 315211, China;

[2]National Climate Centre, China Meteorological Administration, Beijing 100081, China;
[3]Potsdam Institute for Climate Impact Research, Potsdam, Germany;
[4]School of Hydropower and Information Engineering, Huazhong University of Science and Technology, Wuhan, 430074, China;
[5]Collaborative Innovation Center on Forecast and Evaluation of Meteorological Disasters, Institute for Disaster Risk Management (iDRM), School of Geographical Science, Nanjing University of Information Science & Technology, Nanjing 210044, China;
[6]Bureau of Hydrology, Changjiang River Water Resources Commission, Wuhan, 430010, China;
[7]State Key Laboratory of Water Resources and Hydropower Engineering Science, Wuhan University, Wuhan, 430072, China

*Correspondence to*: Tong Jiang (jiangtong@nuist.edu.cn)

**Abstract.** The outputs of four Global Climate Models (GFDL-ESM2M, HadGEM2-ES, IPSL-CM5A-LR and MIROC5), which were statistically downscaled and bias corrected, were used to drive four hydrological models (HBV, SWAT, SWIM and VIC) to simulate the daily discharge at the Cuntan hydrological station in the upper Yangtze River from 1861 to 2299. As the performances of hydrological models in various climate conditions could be different, the models were first calibrated 20 in the period from 1979 to 1990. Then, the models were validated in the wet period, 1967 - 1978, and in the dry period, 1991 - 2002. A multi-objective automatic calibration programme using a univariate search technique was applied to find the optimal parameter sets for each of the four hydrological models. The Nash-Sutcliffe efficiency (NSE) of daily discharge and the weighted least squares function (WLS) of extreme discharge events, represented by high flow (Q10) and low flow (Q90), were included in the objective functions of the parameterization process. In addition, the simulated evapotranspiration results 25 were compared with evapotranspiration data from the GLEAM project for the upper Yangtze basin. For evaluating the performances of the hydrological models, the NSE, modified Kling-Gupta efficiency (KGE), ratio of the root mean square error to the standard deviation of the measured data (RSR) and Pearson's correlation coefficient (r) were used. The four hydrological models showed good performance in the calibration and validation periods. In this study, the daily runoff was simulated for the upper Yangtze River under the preindustrial control (piControl) scenario without anthropogenic climate 30 change, from 1861 - 2299, for the historical period 1861 - 2005, and under the RCP2.6, RCP4.5, RCP6.0 and RCP8.5 scenarios in the period from 2006 to 2299. The long-term daily discharge datasets for the upper Yangtze River provide streamflow trends in the future and clues regarding to what extent human-induced climate change could impact streamflow. The datasets are available at the https://doi.org/10.4121/uuid:8658b22a-8f98-4043-9f8f-d77684d58cbc (Gao et al., 2019).



# 1 Introduction

With the progress of industrialization, global warming has been escalating. Global warming alters global water circulation processes and could significantly influence the sustainability of the social economy (Jung et al., 2011). The variation in water resource availability in the context of global warming has become the focus of hydrological research (Su et al., 2015; Stagl et al., 2016; Raman et al., 2018; Maisa et al., 2019). The long-term accurate daily runoff sequences are crucial for an in-depth understanding of the changes in global water resources and are needed for subsequent research. However, runoff is commonly monitored for only short observational periods in most river basins.

To date, the sedimentological method, hydrological field survey method, and recorded historical documents have been used to develop long-term runoff series (Longfield et al., 2018). However, the low temporal resolution and insufficient estimation accuracy of these methods and resources can hardly meet the demands of practical applications. The observed climatic variables and outputs of Global Climate Models (GCMs) and Regional Climate Models (RCMs) have often been used to drive hydrological models to evaluate changes in runoff in the context of climate change (Braud et al., 2010; Chen et al., 2017; Su et al., 2017; Dahl, 2018; Seneviratne et al., 2018). However, there is a lack of research on the quantitative estimation of long-term runoff for periods longer than 400 years, especially under scenarios without anthropogenic climate change (Meaurio, 2017).

The Yangtze River is the longest river in China. It originates on the Tibetan Plateau and enters the East China Sea after flowing through 11 provinces. With a large topographic gradient and substantial mean annual water supply of approximately 10,000 $m^3s^{-1}$, the upper Yangtze River is rich in hydropower resources but is subject to destructive flash floods. The Yangtze River basin has the longest hydrological observations in China, with data provided by the Cuntan hydrological station, which started operating in 1939. This data availability facilitates hydro-meteorological studies in the instrumental period (Su et al., 2008; Wang et al., 2008; Su et al., 2017). As changes in runoff at the Cuntan station directly influence the water supply of the Three Gorges Reservoir, establishing a long-term runoff time series at the Cuntan station could support the development of hydraulic management strategies in the upper Yangtze River. Therefore, the main aim of this study was to simulate daily discharge at the Cuntan hydrological station in the upper Yangtze River in the period from 1861 - 2299 using available climate model outputs.

The outputs of four downscaled GCMs (GFDL-ESM2M, HadGEM2-ES, IPSL-CM5A-LR, and MIROC5) and four hydrological models (HBV, SWAT, SWIM and VIC) were utilized to simulate runoff at the Cuntan station. The climate forcing comprised (a) scenarios with anthropogenic climate change for the period 1861 - 2299, which was subdivided into the historical period (1861 - 2005) and the future period (2006 - 2299) under different Representative Concentration Pathway (RCP) scenarios, and (b) the preindustrial control (piControl) scenario without human-induced climate change for the period 1861 - 2299, which was used as a reference to detect the influence of anthropogenic climate change on discharge in the upper Yangtze River.



## 2 Study Area

With a catchment area of approximately 860,000 km$^2$, the mean annual runoff at the Cuntan hydrological station (29 ° 37 ′ N, 106 ° 36 ′ E) in the upper Yangtze River is 352.7 billion m$^3$, and the maximum peak discharge is 73,800 m$^3$s$^{-1}$ (in 1945) for the period of instrumental measurements beginning in 1939. Influenced by the East Asian subtropical monsoon and a complex topography, the annual mean temperature exhibited an obvious increasing trend in the upper Yangtze from 11.4 °C (1950s) to 12.4 °C (2010s), with a slope of approximately 0.2 °C / 10 a. The annual precipitation decreased from 900 mm (1950s) to 845 mm (2010s), with a slope of -11 mm / 10 a.

## 3 Data and Methods

### 3.1 Climate scenarios

The outputs of the GCMs (GFDL-ESM2M, HadGEM2-ES, IPSL-CM5A-LR, and MIROC5) were statistically downscaled and bias corrected on a regular 0.5 × 0.5 ° resolution grid using a first-order conservative remapping scheme (Frieler et al., 2017; Lange, 2018). The GFDL model was developed by the Geophysical Fluid Dynamics Laboratory, Princeton University, USA, and all its integrations (approximately 100 in total), including GFDL-ESM2M and GFDL-ESM2G, were completed for the Coupled Model Intercomparison Project Phase 5 (CMIP5) protocol (Taylor et al., 2012). HadGEM2-ES is a coupled earth system model that was developed by the Met Office Hadley Centre, UK, for the CMIP5 centennial simulations (Jones et al., 2011). The IPSL-CM5A-LR model was developed by the Institute Pierre Simon Laplace, France, and the model was built around a physical core that includes atmosphere, land surface, ocean and sea ice components (Dufresne et al., 2013). MIROC5 is a new version of the atmosphere-ocean GCM that was developed by the Japanese research community (Watanabe et al., 2010).

Due to a lack of long-term homogeneous observational data and the confounding influence of socioeconomic drivers, GCM simulations rarely cover the preindustrial period. In this study, the simulated climate conditions include a piControl scenario, representing a preindustrial climate with a $CO_2$ concentration of 286 ppm, a historical scenario, representing the historical $CO_2$ concentration, and future RCP scenarios, representing various future $CO_2$ concentration pathways. The availability of climate scenarios for the different periods is shown in Table 1 (see also Frieler et al., 2017). Note that not all simulations are available after 2099: from three models for RCP2.6, only from IPSL for RCP4.5 and RCP8.5, and no simulations for RCP6.0.

### 3.2 Observed meteorological and hydrological data

The observed daily meteorological data from 1951 - 2017 from 189 ground-based stations in the upper Yangtze River were quality controlled by considering changes in instrument type, station relocations, and trace biases at the National Meteorological Information Center of the China Meteorological Administration (Ren et al., 2010).



The daily discharge data for the period 1939 - 2012 at the Cuntan station in the upper Yangtze River were derived from the China Hydrological Year book - Yangtze. Data from 1967 - 2002 were used for calibration and validation of the four hydrological models.

With large variations in seasonal and interannual precipitation, the Yangtze River is prone to flooding. Disastrous flood events that occurred in 1870, 1931, 1954 and 1998 should be mentioned. In 1870, the most severe flood since 1153 occurred in the upper Yangtze River, with a flood peak at the Yichang station (downstream of the Cuntan station) of approximately 100,500 $m^3s^{-1}$. The peak flows at the Cuntan and Yichang stations reached 63,600 $m^3s^{-1}$ and 64,600 $m^3s^{-1}$, respectively, during the 1931 flood event and 52,200 $m^3s^{-1}$ and 66,800 $m^3s^{-1}$, respectively, during the 1954 flood event. In 1998, the strongest flood of the 20$^{th}$ century occurred in the Yangtze River, and the peak flow at the Cuntan station reached 68,500 $m^3s^{-1}$.

### 3.3 GLEAM evapotranspiration data

In addition to river discharge data, evapotranspiration data from the Global Land Evaporation Amsterdam Model (GLEAM) for the period 1986 - 2005 that were released by the University of Bristol (Miralles et al., 2011) were used to cross-check the performances of the hydrological models. GLEAM is comprised of four mutually connected units: the Gash interception module, soil module, stress module, and Priestley-Taylor module. The remote sensing data were assimilated to obtain monthly evapotranspiration with a spatial resolution of 0.25°.

### 3.4 Hydrological models

Four hydrological models, HBV (Bergstrom et al., 1973), SWAT (Arnold et al., 1998), SWIM (Krysanova et al., 2005) and VIC (Liang et al., 1994), were used to simulate river discharge at the Cuntan hydrological station. A brief introduction to these four hydrological models is given in Table 2 (see also Hattermann et al., 2017).

### 3.5 Calibration and validation methods

A univariate search technique was used to calibrate the parameters (Lai et al., 2006). The objective functions included the Nash-Sutcliffe efficiency (NSE) of daily discharge (Nash and Sutcliffe, 1970) and the weighted least squares function (WLS) of high flow (Q10) and low flow (Q90). To achieve the maximum NSE and minimum difference between the observed and simulated extremes, the parameter values were changed more than 2,000 times within the ranges of the valid parameter scope.

For evaluating daily hydrograph simulations (Moriasi et al., 2007), the ratio of the root mean square error to the standard deviation of measured data (RSR) is recommended. In addition, the Kling-Gupta efficiency (KGE) was developed to provide diagnostic insights into the model performance by decomposing the NSE into three components: correlation, bias and variability (Gupta et al., 2009). In this study, four criteria, the NSE, RSR, Pearson's correlation coefficient (r) and KGE, were applied to the daily time series to evaluate the performances of the hydrological models (Table 3).





### 3.6 Geospatial information

A digital elevation model (DEM) with a resolution of 90 m from the Shuttle Radar Topography Mission database was used in this study. The soil property data were obtained from the Harmonized World Soil Database of the Food and Agriculture Organization of the United Nations (http://www.fao.org/), and the spatial distribution of soil types (1:1,000,000) was taken from the Institute of Soil Science of the Chinese Academy of Sciences (CAS). A land use map (1:1,000,000) for 1990 was provided by the Data Center for Resource and Environmental Sciences of the CAS, and this land use map was applied in the piControl, historical and RCP scenario periods (see discussion in Section 5).

## 4 Results

### 4.1 Climate change in the upper Yangtze basin

Compared to that in the piControl scenario, the annual mean temperature in the historical period, 1861-2005, was slightly higher until the 1980s and showed a notable increase in the period 1986 - 2005 (Fig. 2) according to the ensemble mean of four GCMs. The annual mean temperature in 1986 - 2005 was 0.49 °C higher than that in the period 1861 - 1900 in the upper Yangtze basin, which was lower than the global average of 0.61 °C in the same period. Compared to the piControl scenario, under the RCP scenarios, the annual mean temperature was projected to increase significantly in the 21$^{st}$ century, by 1.1 - 6.9 °C. According to climate model projections, after 2100, the surface air temperature will remain stable under RCP2.6 and increase only slightly under RCP4.5, but a significant increase in temperature will continue under the RCP8.5 scenario, with an increase up to 13.5 °C compared to that in the piControl scenario (Fig. 2a). The visible abrupt changes in temperature in the year 2100 under RCP4.5 and RCP8.5 in Fig. 2a are due to the fact that only the IPSL model runs were available after 2100 for these scenarios.

The long-term average seasonal dynamics of temperature were represented by a single-peak curve, and July was the month with the highest temperature under both the RCP scenarios and the piControl scenario. In the period 1861-2005, the seasonal patterns of temperature were very similar for two scenarios, the piControl and historical scenarios (Fig. 2b). However, differences in the monthly temperatures between the RCP scenarios and piControl scenario become apparent with time (Fig. 2c-d). Taking the temperature in July as an example, the differences between the two scenarios were approximately 1.9-3.2 °C in the 21$^{st}$ century and 1.7-12 °C in the period 2100 - 2299 (Fig. 2c-d).

Compared with the precipitation in the piControl scenario, which had no trend, the annual precipitation in the historical scenario showed a negative trend in the upper Yangtze basin, with an obvious decrease in the period 1986 - 2005 of approximately 7 % (60 mm). Under the RCP scenarios, the annual precipitation was projected to increase in the 21$^{st}$ century by up to 25.4 % compared to the precipitation simulated in the piControl scenario. The increase in precipitation tends to be stable, but the variation amplifies beginning in 2100. Especially under the RCP8.5 scenario, a wide range of fluctuations was





projected with a variance as high as 86.1, which is 60.5 % higher than that in the piControl scenario (Fig. 3a).

The long-term average seasonal precipitation was represented by a single-peak curve, with the highest precipitation in July and the lowest precipitation in December and January. The differences in the long-term average seasonal precipitation under the RCP scenarios and the piControl scenario were projected to be from -1.9 - 1.3 % before 2100 but would grow to -5.4 -

2.2 % in the period 2100 - 2299 (Fig. 3b-d).

### 4.2 Calibration and validation of the hydrological models

Fig. 4 shows the annual precipitation and runoff observed in the upper Yangtze basin in the period 1951 - 2012. The mean observed precipitation and runoff depth were approximately 965 mm and 437 mm, respectively, in the period 1951 - 1986 and decreased by 7 % and 5 % to 895 mm and 415 mm, respectively, in the period 1987 - 2012. As shown in Fig. 4,

1986/1987 could be considered a turning point; the climate conditions become drier after 1987. Therefore, the period 1979 - 1990, which included years with both wet and dry spells, was chosen as the calibration period. Then, the models were validated in two periods without changing the parameters found during the calibration: the wet spell, 1967 - 1978, and the dry spell, 1991 - 2002.

Based on the NSE, RSR and r values, all four hydrological models performed quite well in both the calibration and

validation periods for the simulations of daily discharge at the Cuntan station. In particular, the NSE values of all models exceeded 0.75 in the calibration period and 0.7 in the validation periods (Table 4). The KGE values were above the threshold in the calibration period for all models, but the values were lower in the validation period for the SWIM and VIC models.

The four hydrological models could also properly simulate the high flows represented by Q10 (Fig. 5). In addition, several of the severe observed floods that were mentioned previously were reproduced quite well by the simulated data, namely:

extreme flood event of approximately 36,000 $m^3s^{-1}$ (Fig. 5c) occurred in 1998 during the dry period, and extreme flood event of approximately 32,000 $m^3s^{-1}$ (Fig. 5b) occurred in 1974 during the wet period. The peak flows of simulated discharge in the 1930s, 1950s and 1990s were 64,300 $m^3s^{-1}$, 53,900 $m^3s^{-1}$ and 60,700 $m^3s^{-1}$, respectively, deviating by less than 10 % from the recorded peaks.

To further validate the hydrological models, the discharge simulated in another thirty-year historical period (1939 - 1968)

was compared with the observed data on the monthly scale (Fig. 6). It is visible that SWAT, SWIM and VIC underestimate low flow, and high flow was underestimated in some of the wet years by all hydrological models. But, all four hydrological models could reproduce the monthly river flow quite satisfactorily, with NSE values of 0.79 - 0.84 and $r$ values of 0.91 - 0.92.

In addition, the evapotranspiration outputs of the four hydrological models were compared with the GLEAM

evapotranspiration output (see Section 3.3) in the period 1986 - 2005. The long-term average annual evapotranspiration simulated by the hydrological models for the upper Yangtze basin was 477 mm (range: 426 - 523 mm), which is consistent with the results from GLEAM (466 mm). The spatial patterns of the gridded evapotranspiration outputs of the VIC model and GLEAM are similar (Fig. 7): both models show low values in the north-western region but high values in the

south-eastern region of the upper Yangtze basin. For the other three models, which have spatial disaggregation into sub-basins and not in grid cells, the comparison of spatial patterns of evapotranspiration with the GLEAM output is not shown.

### 4.3 Simulation of daily discharge from 1861 - 2299

The simulated discharge time series under the piControl scenario and scenarios with anthropogenic climate change effects are plotted for the whole period 1861 - 2299 in Fig. 8a-b. In the period 1861 - 2005, the annual mean discharge at the Cuntan station had a slightly decreasing trend, which is similar to the precipitation trend (Fig. 3), which became visible in the late 20th century. Under the RCP scenarios, the annual mean discharge in the upper Yangtze River shows a significant positive trend until 2100, with increasing variation by the end of the 21st century. Beginning in 2100, the annual mean discharge has no significant changes under RCP2.6 and RCP4.5, but a rapid decline in discharge is projected under the RCP8.5 scenario (driven by the IPCL model only).

Higher return levels of daily maximum discharge were projected in the period 2070 - 2099 compared to those in the period 2170 - 2199 (Fig. 8c-d). Generally, the higher the emission scenario, the larger the return level is, with the exception of RCP6.0. When the model projections are taken as a whole, high discharge in the upper Yangtze River shows an increasing trend in the 21st century, turning into a decreasing trend in the 22nd century.

We also compared the changes under the climate warming scenarios in the whole future period 2006 - 2299 to those in the piControl scenario. According to the simulation results of the four hydrological models, the mean annual discharge in the piControl scenario is 11,517 $m^3 s^{-1}$. Relative to that in the piControl scenario, the mean discharge is projected to decrease by 1.7-13.3 % under the RCP scenarios in the period 2006 - 2299 (see Table 5). This result indicates that anthropogenic climate change will induce a decrease in discharge in the upper Yangtze River, and the decrease would be larger under the higher RCP scenarios.

Regarding extremes, the Q90 discharge was projected to be lower under all RCP scenarios compared to that in the piControl scenario. Additionally, the Q10 discharge would also be slightly lower under the three RCP scenarios (except for RCP4.5) in the period 2006 - 2099 (Table 5), indicating an alleviation of flood risks but an aggravation of droughts in the future under global warming.

Regarding discharge variation, both the standard deviation and the coefficient of variation are higher under the RCP scenarios than under the piControl scenario (Table 5), which means that the discharge variation range would increase with the intensification of human-induced climate change.

### 4.4 Data availability

The current study produced the daily discharge time series for the upper Yangtze River (Cuntan gauge station) in the period 1861 - 2299 under scenarios with and without anthropogenic climate change. The river discharge was simulated by four

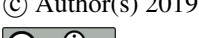



hydrological models, HBV, SWAT, SWIM, and VIC driven by four downscaled and bias-corrected GCMs (GFDL-ESM2M, HadGEM2-ES, IPSL-CM5A-LR and MIROC5), the datasets are available at https://doi.org/10.4121/uuid:8658b22a-8f98-4043-9f8f-d77684d58cbc (Gao et al., 2019).

(1) Scenario without anthropogenic climate change (piControl):

A total of 16 sequences of daily discharge at the Cuntan hydrological station in the upper Yangtze River are the outputs of the four hydrological models that were driven by the four GCMs in the period 1861 - 2299.

(2) Scenarios with anthropogenic climate change:

Historical period: A total of 16 sequences of daily discharge at the Cuntan station in the upper Yangtze River are the outputs of the four hydrological models that were driven by the four GCMs in the period 1861 - 2005.

RCP2.6 scenario: a total of 16 sequences of daily runoff at the Cuntan station in the upper Yangtze River are the outputs of the four hydrological models that were driven by the four GCMs in the period 2006 - 2299 (for GFDL-ESM2M, the sequences are for the period 2006 - 2099).

RCP4.5 scenario: a total of 16 sequences of daily discharge at the Cuntan station in the upper Yangtze River are the outputs of the four hydrological models that were driven by the four GCMs in the period 2006 - 2099 (for IPSL-CM5A-LR, the sequences are for the period 2006 - 2299).

RCP6.0 scenario: a total of 16 sequences of daily discharge at the Cuntan station in the upper Yangtze River are the outputs of the four hydrological models that were driven by the four GCMs in the period 2006 - 2099.

RCP8.5 scenario: a total of 16 sequences of daily discharge at the Cuntan station in the upper Yangtze River are the outputs of the four hydrological models that were driven by the four GCMs in the period 2006 - 2099 (for IPSL-CM5A-LR, the sequences are for the period 2006 - 2299).

## 5 Conclusions and Discussion

Using four GCMs (GFDL-ESM2M, HadGEM2-ES, IPSL-CM5A-LR and MIROC5), changes in temperature and precipitation in the upper Yangtze River basin were analysed from 1861 to the end of 23[th] century under conditions with anthropogenic climate change (the four RCP scenarios) and for a scenario without human-induced climate change (abbreviated as the piControl scenario), and the scenarios were compared. The discharge at the Cuntan station in the period 1861 - 2299 was simulated by four hydrological models (HBV, SWAT, SWIM and VIC) that were driven by the four GCMs, and changes in discharge in a warming world were compared with those in the piControl scenario.

To ensure the reliability of the simulated runoff data, a multi-objective automatic calibration programme using a univariate search technique was applied to obtain the optimal parameter sets for each hydrological model. For the objective functions, the daily discharge and indicators of high and low flows were considered. For the calibration, four criteria, including the NSE, KGE, RSR and r, were used to evaluate the simulation abilities of the hydrological models. To ensure the models' ability to satisfactorily represent discharge under different climate conditions, the hydrological models were additionally





validated in dry and wet periods. In addition, a cross-validation method was applied by comparing the evapotranspiration outputs simulated by the hydrological models with the remote-sensing-based evapotranspiration dataset from the GLEAM.

The results showed that the four hydrological models had good performance in the calibration period and in the both dry and wet periods. Previous studies have also shown that the HBV, SWAT and VIC hydrological models could be applied to the

Cuntan station in the upper Yangtze River after calibration (Huang et al., 2016;Su et al., 2017; Chen et al., 2017). Moreover, the simulated extreme peak values in the 1930s, 1950s and 1990s were also in good agreement with the historical documented records of the catastrophic floods in the Yangtze River.

Although the simulation results were tested and validated with several criteria, there are still uncertainties that could influence the outputs. These uncertainties are associated with the input data (e.g., land use data), downscaling of the GCMs

and setting the climatic scenarios, the model calibration procedure, and water management (Gerhard, et al., 2018). First, as no dynamic land use data were available for the historical period before the 1980s and for the future, a static land use for 1990 was used for simulating river discharge before (including the piControl period) and after the industrial revolution (historical and RCP scenarios). Second, though the most up-to-date climate scenarios were used in this study, downscaling of the GCMs and setting the climate scenarios still contributed to the uncertainty in the hydrological simulation results. Third,

the hydrological models were parameterized using the automatic calibration programme. The parameter effects and model applicability were assessed according to the NSE, KGE, and RSR criteria. However, due to equifinality, there could be other parameter sets that would result in a similarly good model performance. Actually, the combination of parameters and not the choice of individual parameters ultimately influences the result (Cheng et al., 2014). There is a lack of analyses on the effects of different parameter combinations in this study, and the uncertainty related to specific parameters in the models needs to be

analysed further. Fourth, since the 1990s, human interferences have escalated in the upper Yangtze River. The construction of dikes and reservoirs may alter the timing and volume of peak discharge and low flow. The effects of human interferences were not considered in the modelling, which also might bias the simulation results.

The datasets produced in our study are the only available long-term and relatively high-precision discharge sequences for the upper Yangtze River, which includes 16 combinations of outputs of four hydrological models that were driven by four GCM

simulations. The simulations of river discharge under the RCP scenarios with anthropogenic climate change and under the piControl scenario without human-induced climate change could provide support for research on climate change and climate change impacts in the upper Yangtze River basin in the period 1861-2299. Additionally, the simulations also provide clues regarding the extent to which human-induced climate change may impact streamflow in the upper Yangtze River.

**Competing interests**

The authors declare that they have no conflict of interest.



## Acknowledgements

This study was jointly supported by the National Key Research and Development Program of China MOST (2018FY10050001), the National Natural Science Foundation of China (41871024), the High-level Talent Recruitment Program of Ningbo University and the cooperation project between the Natural Science Foundation of China and the Pakistan Science Foundation (41661144027). The authors would like to thank the ISI-MIP project for providing the climate data that was used in this study.

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

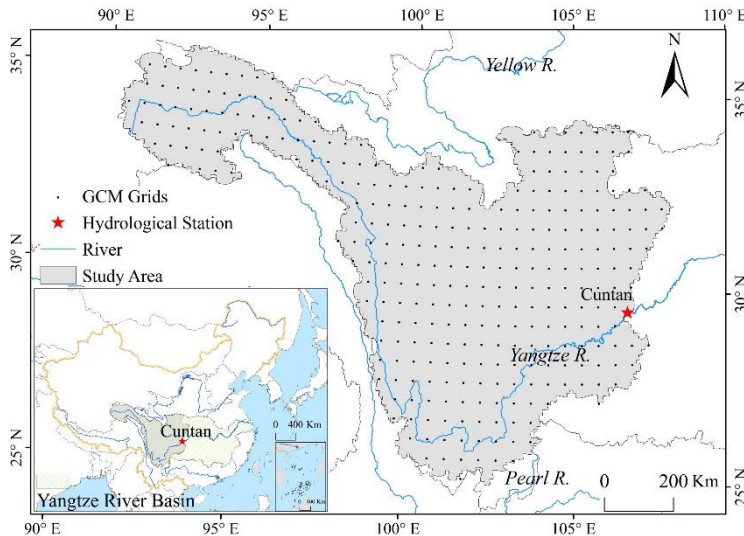

**Figure 1: Location of the Cuntan hydrological station and the GCM grids in the upper Yangtze River basin**



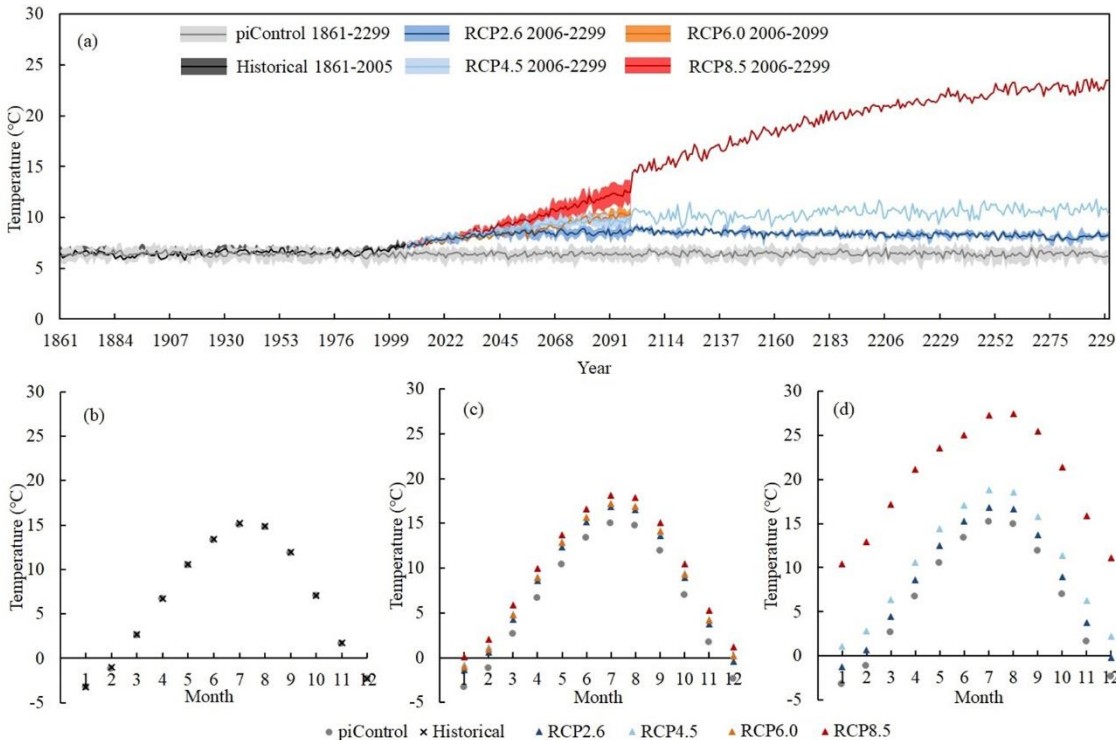

**Figure 2: Interannual (a) and long-term average seasonal (b-d) dynamics of the surface air temperature in the upper Yangtze basin: comparison of the piControl scenario with the historical and anthropogenic climate change RCP scenarios (periods: a: 1861-2299; b: 1861-2005; c: 2006-2099; and d: 2100-2299)**



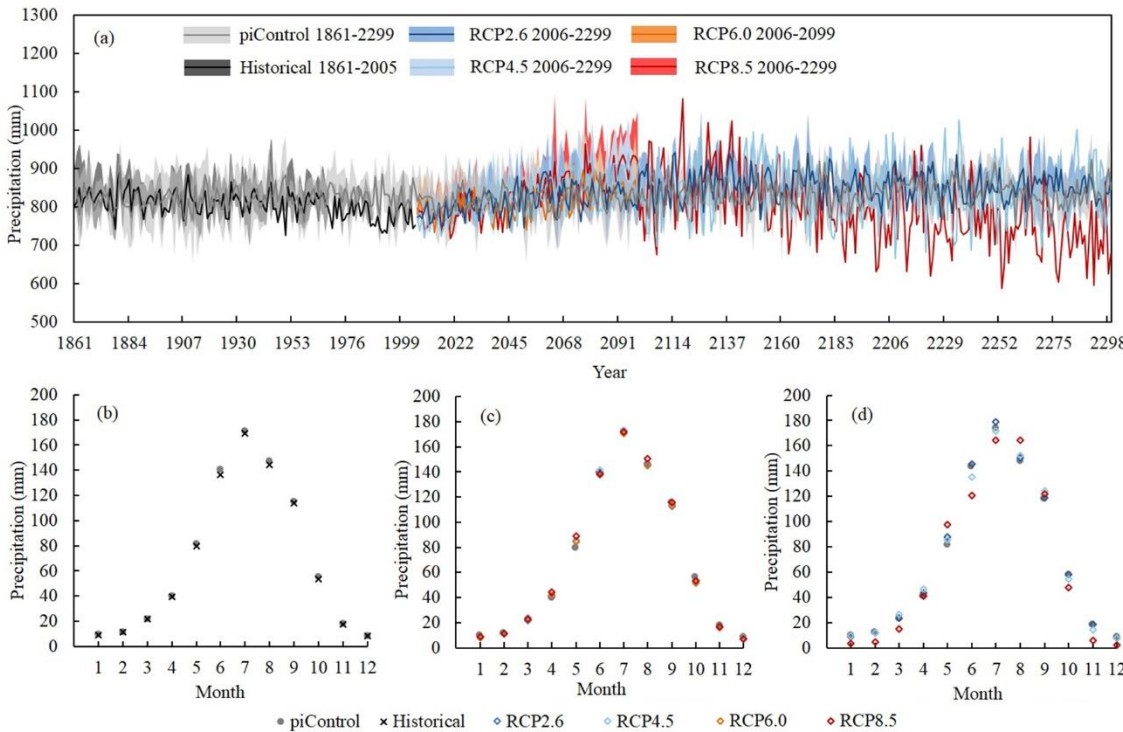

**Figure 3: Annual (a) and long-term average seasonal (b-d) dynamics of precipitation in the upper Yangtze basin: comparison of the piControl scenario with the historical and anthropogenic climate change RCP scenarios (periods: a: 1861-2299; b: 1861-2005; c: 2006-2099; and d: 2100-2299)**





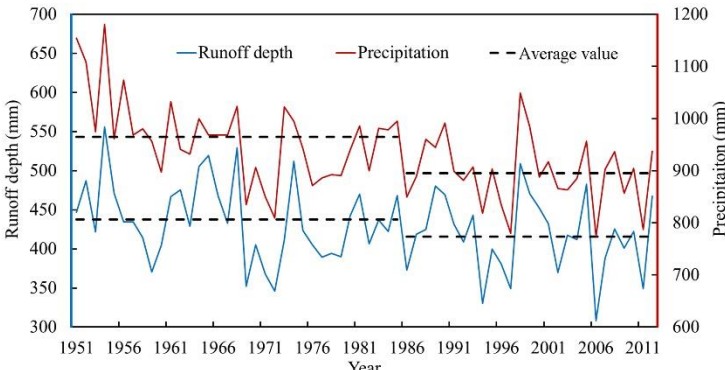

**Figure 4: Annual precipitation and runoff depth observed in the upper Yangtze River basin in the period 1951 - 2012**



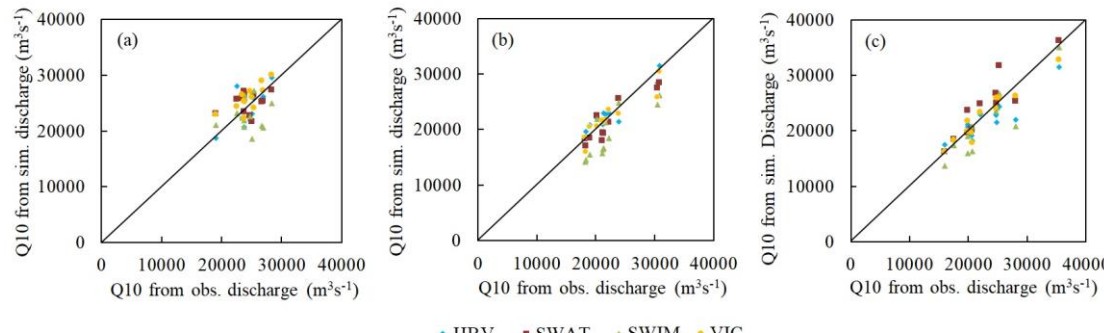

**Figure 5: Comparison of the Q10 values based on the simulated and observed discharge data at the Cuntan station in the calibration period, 1979-1990, (a) and validation periods, 1967-1978 (b) and 1991-2002 (c)**

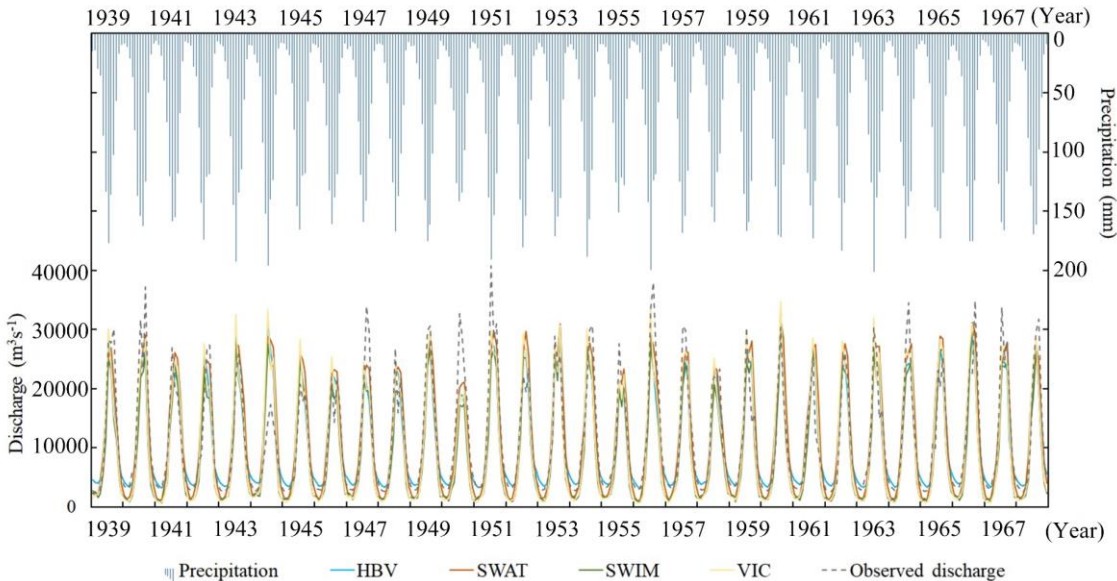

**Figure 6: Observed and simulated monthly discharge and precipitation at the Cuntan station in the upper Yangtze basin in the period 1939 - 1968**



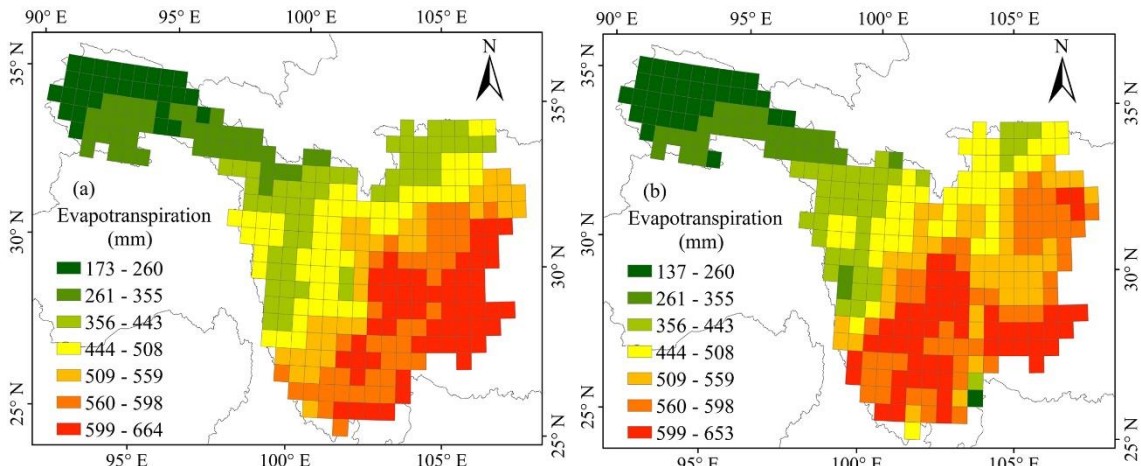

**Figure 7: Spatial distribution of the mean annual evapotranspiration in the upper Yangtze River basin in the period 1986-2005 based on outputs from the hydrological model VIC (a) and GLEAM (b)**



**Figure 8: The annual mean discharge under the piControl scenario and scenarios with anthropogenic climate change effects simulated by the four hydrological models (SWIM, SWAT, HBV, and VIC) (a-b) and the return periods of daily maximum discharge (c-d) at the Cuntan station; comparison of the piControl scenario with the anthropogenic climate change RCP scenarios**

10 **(periods: a-b: 1861-2299; c: 2070-2099; and d: 2170-2199)**



**Table 1: Availability of climate scenarios from four climate models for different periods**

| Climate scenario | $CO_2$ concentration | GFDL-ESM2M | HadGEM2-ES | IPSL-CM5A-LR | MIROC5 |
|---|---|---|---|---|---|
| piControl | 286 ppm | 1861-2099 | 1861-2299 | 1861-2299 | 1861-2299 |
| Historical | Recorded $CO_2$ | 1861-2005 | 1861-2005 | 1861-2005 | 1861-2005 |
| Future | RCP2.6 | 2006-2099 | 2006-2299 | 2006-2299 | 2006-2299 |
| | RCP4.5 | 2006-2099 | 2006-2099 | 2006-2299 | 2006-2099 |
| | RCP6.0 | 2006-2099 | 2006-2099 | 2006-2099 | 2006-2099 |
| | RCP8.5 | 2006-2099 | 2006-2099 | 2006-2299 | 2006-2099 |





**Table 2: Short description of the four hydrological models**

| Model | Institution | Spatial disaggregation | Representation of soils | Representation of vegetation | Routing method |
|---|---|---|---|---|---|
| HBV | Swedish Meteorological and Hydrological Institution | Sub-basins, 10 elevation zones & land use classes | 1 soil layer, 2 soil parameters | Fixed monthly plant characteristics | A simple time-lag method |
| SWAT | United States Department of Agriculture | Sub-basins and hydrological response units | Up to 10 soil layers, 11 soil parameters | A simplified EPIC approach | Muskingum method |
| SWIM | The Potsdam Institute for Climate Impact Research, based on the SWAT and MATSALU models | Sub-basins and hydrotopes | Up to 10 soil layers, 11 soil parameters | A simplified EPIC approach | Muskingum method, reservoirs and irrigation |
| VIC | University of Washington, University of California, and Princeton University | Grid of large and uniform cells with sub-grid heterogeneity | 3 soil layers, 19 parameters | Fixed monthly plant characteristics | Linearized St. Venant's equations |





**Table 3: Evaluation criteria for testing performance of hydrological models**

| Criterion | Formula | Range | Ideal value | Notation |
|---|---|---|---|---|
| Nash-Sutcliffe efficiency (NSE) | $1 - \dfrac{\sum_{t=1}^{N}(Q_{s,t} - Q_{o,t})^2}{\sum_{t=1}^{N}(Q_{o,t} - \bar{Q}_o)^2}$ | $(-\infty, 1)$ | 1 | $Q_s$: simulated discharge; $Q_o$: observed discharge; $\bar{Q}_o$: mean of observed |
| Ratio of the root mean square error and the standard deviation of observation (RSR) | $\dfrac{\sqrt{\sum_{t=1}^{N}(Q_{o,t} - Q_{s,t})^2}}{\sqrt{\sum_{t=1}^{N}(Q_{o,t} - \bar{Q}_o)^2}}$ | $(0, +\infty)$ | 0 | discharge; $\bar{Q}_s$: mean of simulated discharge; |
| Pearson's correlation coefficient ($r$) | $\dfrac{\sum_{t=1}^{N}(Q_{s,t} - \bar{Q}_s)(Q_{o,t} - \bar{Q}_o)}{\sqrt{\sum_{t=1}^{N}(Q_{s,t} - \bar{Q}_s)^2} - \sqrt{\sum_{t=1}^{N}(Q_{o,t} - \bar{Q}_o)^2}}$ | $(-1, 1)$ | 1 | $t$: sequence of the discharge series; N: number of time steps; |
| Modified Kling-Gupta efficiency (KGE) | $1 - \sqrt{(\alpha - 1)^2 + (\beta - 1)^2 + (r - 1)^2}$ | $(-\infty, 1)$ | 1 | $\alpha$: ratio between the standard deviations of the simulated and observed data; $\beta$: ratio between the mean simulated and mean observed discharge |



**Table 4: Criteria of fit of the four hydrological models in the calibration period and in the wet and dry validation periods**

| Criterion | Threshold | Calibration/validation | HBV | SWAT | SWIM | VIC |
|---|---|---|---|---|---|---|
|  |  | 1979-1990 | 0.86 | 0.81 | 0.75 | 0.89 |
| NSE | >=0.7 | 1967-1978 (wet period) | 0.86 | 0.79 | 0.7 | 0.88 |
|  |  | 1991-2002 (dry period) | 0.86 | 0.81 | 0.75 | 0.89 |
|  |  | 1979 - 1990 | 0.39 | 0.43 | 0.50 | 0.33 |
| RSR | <=0.6 | 1967-1978 (wet period) | 0.38 | 0.46 | 0.55 | 0.34 |
|  |  | 1991-2002 (dry period) | 0.36 | 0.42 | 0.48 | 0.32 |
|  |  | 1979-1990 | 0.92 | 0.91 | 0.91 | 0.97 |
| $r$ | >=0.9 | 1967-1978 (wet period) | 0.92 | 0.90 | 0.89 | 0.96 |
|  |  | 1991-2002 (dry period) | 0.94 | 0.92 | 0.93 | 0.97 |
|  |  | 1979-1990 | 0.87 | 0.9 | 0.7 | 0.71 |
| KGE | >=0.7 | 1967-1978 (wet period) | 0.90 | 0.88 | 0.65 | 0.69 |
|  |  | 1991-2002 (dry period) | 0.85 | 0.89 | 0.56 | 0.68 |



**Table 5: Comparison of changes in the mean annual discharge, Q10 and Q90 in the period 2006-2299 under the scenarios of anthropogenic climate change and the piControl scenario**

| Dataset | Rate of change (Mean) (%) | Rate of change (Q10) (%) | Rate of change (Q90) (%) | Standard deviation | Coefficient of variation |
|---|---|---|---|---|---|
| piControl | - | - | - | 980.4 | 0.08 |
| RCP2.6 | -0.92 | -0.49 | -3.8 | 1146.2 | 0.10 |
| RCP4.5 | -4.7 | 5.2 | -11.9 | 1819.6 | 0.17 |
| RCP6.0 | -7.7 | -7.3 | -7.9 | 965.9 | 0.09 |
| RCP8.5 | -18.2 | -2.9 | -30.6 | 2347.9 | 0.25 |