# Peer review of "A 439-year simulated daily discharge dataset (1861-2299) for the upper Yangtze River, China"

_Earth System Science Data, 2019_

## Referee Comment (RC1) · Anonymous Referee #1 · 15 Sep 2019

**General Comments**

*Overall quality of the paper*

Firstly, thanks the authors for this manuscript that includes a good hydrological modeling effort by driven Global Climate Models (GCMs) under climate change scenarios. The paper presents multi-hydrological modeling by inputting GCMs data at daily time scale and impacts of climate change on Yangtze River, China.

The methods of the paper don't include scientific newness. The methods and the materials aren't described in sufficient detail, there are some deficiencies which are remarked at related points. Despite these, it is generally clear, well-structured and well-written but the language could be improved some in the sense of fluency and clarity. And, the manuscript is suitable in terms of themes of ESSD journal. I think it could be accepted after revision based on the review.

Advantages of the manuscript

- Multi-hydrological model approach
- GCMs under climate change scenarios
- Internal consistency evaluation of hydrological models by using GLEAM evapotranspiration data

Disadvantages of the manuscript

- The lack of multi-site (which have different geo-climatic characteristics) comparison)
- Use of 1990 year land use data for particularly future scenarios
- The deficiency of comparison of GCMs based on observed data
- The poor comparison of hydrological models based on observed data

**Specific Comments**

The title could be edited more suitably for the main purpose of the paper. In the title, "hydrological modeling and GCMs" expressions could be used.

In the abstract; the objective , the original value and the widespread impact of the study could be expressed better.

The introduction could be handled for meaning unity and logical continuity.

In the section of study area, Fig. 1 must be pointed out and river basin characteristics (elevation, slope, aspect, land cover, soil type, etc.) must be given shortly to understand the hydrological cycle of the basin.

In the methods, a flowchart included all processes could be given for the sake of clarity. Moreover, GIS and remote sensing techniques used could be mentioned shortly in this section.

In section 3.1, some information about downscaling, bias correction and data assimilation processes (details, techniques, etc.) of GCMs should be given. And it should be explained that how GCMs grid data was inputted into the hydrological models (as lumped data?). In addition, more details about GCMs (model structures, etc.) and RCP scenarios should be presented.

In section 3.2, please provide details (precipitation, temperature, etc.) of observed daily meteorological data and explain how the data of 189 ground-based stations was inputted into the hydrological models (mean or spatial interpolation/extrapolation of the data?). Furthermore, please give reference/references of flood events (1870 and 1931) because of measurements beginning in 1939.

In section 3.3, please give more details about GLEAM and the assimilation technique. And please present the comparison results of GLEAM and 4 hydrological models for internal consistency of the models in the results section.

In section 3.4, please present input and output data of the models, model structures (loss and transform methods, soil moisture accounting, snow melting-accumulation, evapotranspiration, groundwater, canopy interception etc.) and parameter sets.

In section 3.5, please indicate how many parameters were calibrated for each model and what is the range for each parameter?

In section 3.6, please clarify the problem of "is the use of 1990 year land use convenient for particularly future scenarios?". In conclusions and discussion section, it is stated that 1990 year land use is one of the uncertainties and human interferences have escalated since the 1990s. In my opinion, using 1990 land use data is a big handicap of this study that is not convenient for future modeling. 1990 land use data can be considered suitable for only piControl and historical periods. Modeling of future flows by using the land cover of 1990, when the industrial development has not started, will have misleading results, especially in human impact scenarios. The use of 2019 year land cover data would be more appropriate when modeling the future and this will make the results more consistent. I don't accept the pretext of not available by reason of the fact that current land use data can be easily obtained from various sources.

In section 4.1 and 4.3, while determining the trends, was statistical trend analysis applied? Or how were the trends determined? By only visual technique?

In section 4.2, please explain what the model performance threshold is. Is it for good performance? Please give reference/references for it (Table 4). In addition, please comment why SWIM and VIC models had lower performance in the validation period. Moreover please provide Q90 results in Fig. 5 which shows the scatter plots. So, in this figure, the coefficient of determination ($R^2$) of 4 hydrological models should be given for significance of the scatter plots. Likewise, WLS values of extreme conditions should be presented for the calibration period. For this section, finally, please show the mean annual evapotranspiration outputs of the other three models based on sub-basins in Fig. 7 and compare all of the models with GLEAM based on statistical and/or mathematical criteria. It is better to see the all models in

comparison. Even further, VIC and GLEAM could be compared based on grid by means of a confusion matrix.

In competing interests section, please give information of author contribution for each author in favor of justice.

In Fig. 1, please present DEM to get to know the basin better. And please specify how many grid cells are in the basin.

In titles of Fig. 2 and Fig. 3, "seasonal" expressions should be replaced with "monthly".

Although the study is daily flow modeling, the graphical representation in Fig. 6 is given on monthly time scale since the application period is very long. If so, please express this.

In Fig. 8, are the values mean outputs of hydrological models and GCMs (a-b)? And there is a noteworthy decreasing in discharge (d), why? Please comment. And why is "2100-2169" period not available (c-d)?

In Table 4, it is interesting that the performance criteria are so close between the cal-val periods. Please comment what the reason.

**Technical Corrections**

**P3 L6 and L7:** "0.2 $^{o}$C/10$\underline{a}$" and "-11 mm/10$\underline{a}$" $\rightarrow$ Please present as open.

**P4 L27:** "(Moriasi et al., 2007)" must be placed in the end of the sentence.

**P5 L21-L22 and P6 L2-L3:** "seasonal" $\rightarrow$ "monthly"

**P6 L27:** "models could reproduce the monthly river flow …" $\rightarrow$ Please explain which one of "monthly modeling" or "daily modeling and transforming into monthly" was performed.

**P7 L13 and L14:** "with the exception of RCP 6.0" $\rightarrow$ Please comment.

**P7 L14 and L15:** "an increasing trend in the 21$^{st}$ century, turning into a decreasing trend in the 22$^{nd}$ century." $\rightarrow$ Why? Please explain considering the precipitation.

**P7 L18:** "11,517 m$^3$s$^{-1}$" $\rightarrow$ "11,517 m$^3$s$^{-1}$ (363.2 billion m$^3$)". For comparison of observation value stated in the section of study area.

**P7 L23:** "(except for RCP 4.5)" $\rightarrow$ Why? Please interpret.

**P8 L30:** "For the calibration" $\rightarrow$ "For the calibration and the validation periods"

**P9 L1:** "a cross-validation method" $\rightarrow$ Please give info. Leave-one-out or k-fold technique? If leave-one-out technique, it is problem of independence of validation dataset.

**P9 L6 and L7:** "the simulated extreme peak values in the 1930s, 1950s and 1990s were also in good agreement with the historical documented records …" $\rightarrow$ Please add "except 1998 flood" for objectiveness and being non-manipulative. It is specified that the modeled peak flow is 36,000 m$^3$/s (in **P6 L20**) and the observed value is 68,500 m$^3$/s (in **P4 L9**) for the 1998 flood event.

The references in **P12 L5 and P12 L8** must be replaced due to the alphabetical order.

**Recommendations**

- The model performance values for the discharge and the evapotranspiration could be combined to consider the internal consistency of the model. As well in the calibration period, the model could be calibrated according to both the discharge and the evapotranspiration (multi-criteria calibration).
- The hydrological models can be performed in different areas in point of multi-site approach.

**Data Set Control-Evaluation**

In accordance with the aim-scope of the journal, the dataset control process of the review is rather important case. For future reuse and reinterpretation, I checked the quality of the datasets which are available at the relevant web link.

I think there is potential of the data being useful in the future but there are only simulated discharge file (xls) as time series. The observed data (discharge, precipitation, temperature, etc.), the data of GCMs (precipitation, temperature, etc. for pi and RCP scenarios), the evapotranspiration data of 4 hydrological models and GLEAM (as time series), the grid evapotranspiration data (GLEAM and VIC) and the other grid data used (DEM, soil and land use data) must be also presented for both reproducibility of scientific and usefulness of data. In addition, to perform tests for data quality, the above mentioned data sets are necessary. The availability of these datasets are important for usefulness and completeness, too. And at the aim and scope web page of the journal, the expression of "each article should publish as much data as possible" supports the completion of deficiencies.

At the datasets web link page, in Table 3 of README.pdf file, "Future" words for 4 hydrologic models were miswritten.

---

## Referee Comment (RC2) · Anonymous Referee #2 · 30 Sep 2019

Add the word "simulated" or something similar to the title. I know that with the dates it is clear that this is enough an actual 439 year dataset of measured data, but save the reader from momentary excitement at a record being found that goes back that far.

What is the target audience for this paper? I think the goals of the paper need to be more clearly stated.

Who do you envision using this dataset? For what potential purposes?

In the goals for ESSD, does this paper provide "original research data"? I am not a modeler and thus am not up to date on whether any of the methods presented are new or novel.

A lot of sentences in this paper start with a weak dependent clause and then leads

into the independent clause. I've tried to highlight some of them, but the language in the paper could be made stronger by reducing the number of these sentences. It's ok to have a short sentence every now and then, especially if it conveys important information.

Figure 1 is not referenced anywhere in the paper.

Make it clear what the period of daily measured discharge is.

1. Intro

First sentence of the intro should be stronger, don't start with "with".

Line 8 of the intro, don't start with "To date"

Line 17 needs a comma before "but"

Line 23 – "Could support the development. . ."? Is there not already hydraulic management strategies in place?

2. Study Area

Is the temperate trend linked to the East Asian monsoon and topography?

Don't say the temperate trend is obviously increasing, especially without showing a graph. By reporting a slope of "approximately 0.2C/10a" you are implying a linear trend, is this the case?

Are you referring to air temperature or water temperature?

Did precipitation decrease linearly (as implied by the slope)? Does this correlate with the wet period and dry periods previously mentioned?

3. Methods

I will let another reviewer determine whether the models were used appropriately.

3.2.

What do you mean the daily discharge data "were derived"?

Why was data from 1939-1969 not used for calibration and validation of the four hydrological models?

Reorder the first sentence of paragraph 2 on page 4.

Remove language from the next sentence than the disastrous floods should be mentioned.

How do you know that the 1870 flood was the most severe since 1153? What do you mean by "severe"?

The language in this paragraph could be cleaned up and made much tighter.

4.1

Reorder the first sentence

4.2

1986/1987 does not look like a turning point to me.

4.3

Why does the IPCL model only project a rapid decline (should be decrease?) in discharge. Seems like that would be an interesting point to discuss.

Is mean annual discharge really the best way to characterize the discharge with how much it fluctuates throughout the year?

Will you later go into why discharge is projected to decrease? This seems like an interesting conclusion to investigate further.

4.4

What do you mean 16 sequences of daily discharge?

5.

I would like to see discussion and conclusions separated unless this is the journal's recommended style.

Is there no updated land use map since 1990? I would think the effects of human induced change could completely change the results and needs to be incorporated. There has been tremendous growth in the last (almost) 30 years in China.

Figure 3a could be a dynamic figure. Can you make it clearer? If it's not possible in one graph, maybe split it up?

Figure 3 captions needs more information.

Figure 4 – why the two different average value lines? Needs to be discussed in the figure caption. Is this the wet/dry periods?

Figure 5 – include r values

Figure 8 – Does historical data only go to 2005? This needs to be mentioned earlier in the paper. Is there more data available ie through at least 2018 or whenever the analysis was started?

---

## Author Comment (AC1) · 22 Dec 2019

Dear referee,

Thank you very much for reviewing the manuscript and providing comments. Your comments are helpful for improving the manuscript. In the following, we address all comments point-by-point.

*Comment 1: Add the word "simulated" or something similar to the title. I know that with the dates it is clear that this is enough an actual 439 year dataset of measured data, but save the reader from momentary excitement at a record being found that goes back that far. What is the target audience for this paper? I think the goals of the paper need to be more clearly stated.*

**Answer:**

We added the "simulated" before the word "daily" as follow:

"A 439-year simulated daily discharge dataset (1861-2299) for the upper Yangtze River, China"

We added one sentence to abstract:

"The long-term daily discharge dataset can be used in the international context and water management, e.g. in the framework of Inter-Sectoral Impact Model Intercomparison Project (ISIMIP) by providing clue to what extent human-induced climate change could impact streamflow and streamflow trend in future."

*Comment 2: Who do you envision using this dataset? For what potential purposes?*

**Answer:**

In order to respond the simulation protocol of the Inter-Sectoral Impact Model Intercomparison Project (ISIMIP2b: https://www.isimip.org/protocol/#isimip2b). This dataset is used widely for cross-sectoral projections and can also be applied to assess changes in river discharge attributable to anthropogenic climate change.

*Comment 3: In the goals for ESSD, does this paper provide "original research data"? I am not a modeler and thus am not up to date on whether any of the methods presented are new or novel.*

**Answer:**

We provided the outputs simulated by four hydrological models and will upload the original research data and simulated data as much as possible. The novelty of our manuscript is quite clear. We use multiple hydrological models, which were driven by multiple GCMs under scenarios with and without anthropogenic climate change effects, to deduce the longer discharge series over 400 years.

*Comment 4: Figure 1 is not referenced anywhere in the paper*

**Answer:**

We have added a reference to Fig. 1 in the section "Study area": Location of the Cuntan hydrological station, 311 GCM grids, meteorological stations and spatial distribution of the land use and soil types in the upper Yangtze River basin are shown in Fig. 1.

*Comment 5: Make it clear what the period of daily measured discharge is.*

**Answer:**

"The daily discharge record at the Cuntan station in the upper Yangtze River is available for 1970 - 1999 from the China Hydrological Yearbook - Yangtze. The rest of daily record for periods 1939 - 1969 and 2000 - 2012 is collected from the Changjiang Water Resources Commission, Ministry of Water Resources in China." (Section 3.2, page 4, line 11).

**1. Introduction**

*Comment 6: First sentence of the intro should be stronger, don't start with "with".*

**Answer:**

We modified the sentence in the revised paper:
"Global warming is the long-term rise in average temperature of the earth's climate system. Warming temperature alters global water circulation processes and could significantly influence the sustainability of society and economy (Jung et al., 2011). The variation in water resource availability in the context of global warming is acknowledged as a focus of many international research projects (Stagl et al., 2016; Raman et al., 2018; Maisa et al., 2019). The long-term accurate (as much as possible) daily discharge time series are crucial for in-depth understanding of the changes in streamflow, and they are needed for subsequent climate change impact studies. However, discharge is monitored usually only for short observational periods in most river basins."

*Comment 7: Line 8 of the intro, don't start with "To date"*

**Answer:**

We revised the sentence as:
"For generation of the long-term streamflow series, many data mining techniques including the sedimentological method, the hydrological field survey method, and the documentary analysis method can be applied (Longfield et al., 2018). Nevertheless, low temporal resolution and

insufficient accuracy of these estimations can hardly meet the demands of practical and research applications."

*Comment 8: Line 17 needs a comma before "but"*

**Answer:**

We are appreciating your advice and added a comma:
"With a large topographic gradient and substantial water supply of approximately 10,000 $m^3s^{-1}$ on the average, the upper Yangtze River is rich in hydropower resources, but subjected to destructive flash floods."

*Comment 9: Line 23 – "Could support the development..."? Is there not already hydraulic management strategies in place?*

**Answer:**

It can support the development and we changed "could" with "can". Sure, there are hydraulic management strategies in place, but they will be also developed in future. "As changes in streamflow at the Cuntan station directly influence inflow to the Three Gorges Reservoir, establishing long-term discharge series at the Cuntan station can support effective management of hydraulic projects. Besides, the longer discharge series can also provide a possibility to explore impacts of anthropogenic climate change on hydrology for international climate change research community. Therefore, we simulated daily discharge at the Cuntan hydrological station in the upper Yangtze River in the period 1861 - 2299 using available climate model outputs."

**2. Study area**

*Comment 10: Is the temperate trend linked to the East Asian monsoon and topography?*

**Answer:**

As far as we know, spatial and temporal patterns of temperate are linked to the East Asian monsoon and topography. In order to make it clearer, we rewrote the description in revised manuscript as following:
"The upper Yangtze River have complex geomorphic types and broken topography. Mountains and plateaus account for most of the region, hills and plains are few. Influenced by the East Asia subtropical monsoon and a complex topography, climate varies across the basin with annual air temperature and precipitation being high in the southeast but low in the northwest headstream region. According to observational data, the areal averaged annual mean temperature and precipitation are 12.3 ℃ and 1018 mm, respectively, during 1961 - 2017 in the upper Yangtze River basin."

*Comment 11: Don't say the temperate trend is obviously increasing, especially without showing a graph. By reporting a slope of "approximately 0.2C/10a" you are implying a linear trend, is this the case?*

**Answer:**

We tried to exhibit mean air temperature from the observed data, and modified the sentence as: "According to observational data, the areal averaged annual mean temperature and precipitation are 12.3 ℃ and 1018 mm, respectively, during 1961 - 2017 in the upper Yangtze River basin."

*Comment 12: Are you referring to air temperature or water temperature?*

**Answer:**

The mentioned temperature is air temperature. We added "air" before temperature in the revised paper (page 3, line 5):
"Influenced by the East Asia subtropical monsoon and a complex topography, climate varies across the basin with annual air temperature and precipitation being high in the southeast but low in the northwest headstream region."

*Comment 13: Did precipitation decrease linearly (as implied by the slope)? Does this correlate with the wet period and dry periods previously mentioned?*

**Answer:**

It exhibited mean air precipitation from the observation, and no correlation with the wet and dry periods previously mentioned. We tried to deliver some general meteorological information about the upper Yangtze River.

**3. Data and Methods**

*Comment 14: I will let another reviewer determine whether the models were used appropriately. What do you mean the daily discharge data "were derived"?*

**Answer:**

We revise the sentence as:
"The daily discharge record at the Cuntan station in the upper Yangtze River is available for 1970 - 1999 from the China Hydrological Yearbook - Yangtze. The rest of daily record for periods 1939 - 1969 and 2000 - 2012 is collected from the Changjiang Water Resources Commission, Ministry of Water Resources in China."

*Comment 15: Why was data from 1939-1969 not used for calibration and validation of the four hydrological models?*

**Answer:**

The discharge in the upper Yangtze have been observed since 1939, but majority of meteorological stations started to operate at early 1950s. The period 1979 - 1990, which included years with both wet and dry spells, was chosen as the calibration period. Then, the models were validated in two periods without changing the parameters set in the calibration: the wet spell, 1967 - 1978, and the dry spell, 1991 - 2002, following recommendation for model evaluation in reference (Krysanova et al., 2018).

*Comment 16: Reorder the first sentence of paragraph 2 on page 4.*

**Answer:**

We have reordered the first sentence of paragraph 2 on page 4:
"The Yangtze River is prone to be flooded because of large inter- and inner-annual variations of precipitation."

*Comment 17: Remove language from the next sentence than the disastrous floods should be mentioned.*

**Answer:**

We are appreciating your advice and added some reference:
"The most severe flood that can be tracked in the upper Yangtze River occured in 1870, with a flood peak of approximately 100,500 $m^3s^{-1}$ at the Yichang station located downstream of the Cuntan station (Changjiang Water Resources Commission, 2002). The peak flows reached 63,600 $m^3s^{-1}$ and 64,600 $m^3s^{-1}$, respectively, at the Cuntan station and the Yichang station during the 1931 flood, and 52,200 $m^3s^{-1}$ and 66,800 $m^3s^{-1}$, respectively, during the 1954 flood (Hu and Luo, 1992; Luo and Le, 1996). During the strongest flood of the 20th century in the Yangtze River, the peak flow at the Cuntan station reached 68,500 $m^3s^{-1}$ in 1998 (Changjiang Water Resources Commission, 2002)."

*Comment 18: How do you know that the 1870 flood was the most severe since 1153? What do you mean by "severe"?*

**Answer:**

The 1870 flood was recognized as the most disastrous flood events by Changjiang Water Resources Commission according to various historical records. We cited it in the revised paper as "The most severe flood that can be tracked in the upper Yangtze River occurred in 1870, with a flood peak of approximately 100,500 $m^3s^{-1}$ at the Yichang station located downstream of the Cuntan station (Changjiang Water Resources Commission, 2002)." The "severe" is used here because the flood of 1870 reached the highest flood level in hundreds of years in the upper Yangtze river (Changjiang

Water Resources Commission, 2002).

*Comment 19: The language in this paragraph could be cleaned up and made much tighter.*

**Answer:**

We added references to make this paragraph much tighter: "For evaluating daily hydrograph simulation, ratio of the root mean square error to the standard deviation of measured data (RSR) is recommended (Moriasi et al., 2007). In addition, the Kling-Gupta efficiency (KGE) was developed to provide diagnostic insights into the model performance by decomposing the NSE into three components: correlation, bias and variability (Gupta et al., 2009)." We have modified statement throughout the article to make much tighter.

**4.1 Climate change in the upper Yangtze basin**

*Comment 20: Reorder the first sentence*

**Answer:**

Thanks for your suggestion. We have reordered the first sentence:
"According to ensemble mean of four GCMs, annual mean temperature in the upper Yangtze River basin in the period 1986 - 2005 was 0.49 °C higher than that in the period 1861 - 1900, the increase is lower than the global average of 0.61 °C in the same period."

**4.2 Calibration and validation of the hydrological models**

*Comment 21: 1986/1987 does not look like a turning point to me (4.2)*

**Answer:**

As you mentioned, 1986/1987 does not look like a turning point. We have taken the Pettitt test and found that the turning point is in 1989. However, 1986 had comparatively less precipitation and low runoff (Fig. 5) and can be regarded as the start year of being drier. We accept your comment and rewrote the sentence and added one reference:
"Previous study found that 1986/1987 was a change-point in the observational period for south China, with more obvious increase of temperature and decrease of precipitation since then (Thomas et al., 2012)."

[Figure]

Figure 5 Annual precipitation and runoff depth observed in the upper Yangtze River basin in the period 1951 - 2012

**4.3 Simulation of daily discharge from 1861 - 2299**

> ***Comment 22****: Why does the IPSL model only project a rapid decline (should be decrease?) in discharge. Seems like that would be an interesting point to discuss.*

**Answer:**

Yes, it is. The IPSL model has an obvious decrease in precipitation (Fig. 4b: RCP8.5), which was used by hydrological models as an input to simulate the discharge. Therefore, a significant decrease in discharge was projected. The reasons why IPSL model projected such a decrease in precipitation are not known for us.

> ***Comment 23****: Is mean annual discharge really the best way to characterize the discharge with how much it fluctuates throughout the year?*

**Answer:**

One of the optimal methods to evaluate the simulated long-term discharge is to apply the mean value. We added average monthly streamflow to characterize its inter-annual fluctuation, and found a single peak pattern throughout the year (see Fig. 10a-b as follow).

[Figure]

Figure 10 Comparison of monthly mean simulated discharge and return periods of daily maximum discharge at the Cuntan station for 2070 - 2099 (a, c) and 2270 - 2299 (b, d) under RCPs and the piControl scenario

**Comment 24:** *Will you later go into why discharge is projected to decrease? This seems like an interesting conclusion to investigate further.*

**Answer:**

According to our simulation results, the daily simulated discharge will reduce with the decrease of precipitation in the future. It is a good idea to further study the reasons why discharge is projected to decrease, which should be taken into consideration in the future.

**4.4 Data availability**

**Comment 25:** *What do you mean 16 sequences of daily discharge?*

**Answer:**

4 GCMs and 4 different hydrological models are used to simulate river discharge in our study, and each GCM was inputted into 4 hydrological models. Therefore, total of 16 discharge series are outputted.

**5 Summary and conclusions**

> *Comment 26: I would like to see discussion and conclusions separated unless this is the journal's recommended style.*

**Answer:**

There is no specific recommend style in ESSD. Section 5 is a summary about how we obtained this dataset and verified its reliability. Thus, we changed the subtitle "Conclusions and discussion" to "Summary and conclusions".

> *Comment 27: Is there no updated land use map since 1990? I would think the effects of human induced change could completely change the results and needs to be incorporated. There has been tremendous growth in the last (almost) 30 years in China.*

**Answer:**

Actually, influence of human management has aggravated and the land cover changed quite obviously in the last 30 years in China. Therefore, calibration is conducted in a comparatively earlier period of 1979-1990. In order to compare the trend of daily simulated discharge under a changing climatic background and keep the hydrological model stable with high performance in different period, we used the 1990 land use as single geographical data to simulate the daily discharge by controlling the variable as little as possible.

> *Comment 28: Figure 3a could be a dynamic figure. Can you make it clearer? If it's not possible in one graph, maybe split it up? Figure 3 captions needs more information.*

**Answer:**

Fig. 4a and 4b (previous Fig. 3) show the annual dynamics of precipitation for a long period, splitting it into subperiods might not be reasonable.

[Figure]

Figure 4 Annual (a-b) and long-term average monthly (c-e) dynamics of precipitation in the upper Yangtze basin: comparison of the piControl scenario with the historical and anthropogenic climate change RCP scenarios (periods: a: 1861 - 2299; b: 1861 - 2299; c: 1861 - 2005; d: 2006 - 2099; and e: 2100 - 2299)

*Comment 29: Figure 4 – why the two different average value lines? Needs to be discussed in the figure caption. Is this the wet/dry periods?*

**Answer:**

Fig. 5 illustrates the wet/dry periods based on the observed discharge and meteorological data. Two dotted straight lines represent the mean discharge and precipitation, respectively, in the wet and dry period.

**Comment 30:** *Figure 5 – include r values*

**Answer:**

We have redrawn Fig. 5 (now Fig. 6), which compares the simulated and observed Q10, Q90 discharges at the Cuntan station in the calibration and validation periods.

[Figure]

Figure 6 Comparison of the simulated and observed Q10, Q90 percentiles at the Cuntan station in the calibration period 1979 - 1990 (a) and validation period 1967 - 1978 and 1991 - 2002 (b-c)

**Comment 31:** *Figure 8 – Does historical data only go to 2005? This needs to be mentioned earlier in the paper. Is there more data available ie through at least 2018 or whenever the analysis was started?*

**Answer:**

The fourth paragraph in introduction describes the time ranges of GCMs' historical period as defined in the ISIMIP protocol (Frieler et al., 2017). It was defined that historical anthropogenic climate change period is 1861-2005, which can be compared with piControl scenario without human-induced influences and future RCPs, and we followed the suggestions in the protocol.

Best regards,

Chao Gao and co-authors

[revised manuscript text omitted]
}\left(Q_{s,t} - Q_{o,t}\right)^2}{\sum_{t=1}^{N}\left(Q_{o,t} - \bar{Q}_o\right)^2}$ | $(-\infty, 1)$ | 1 | $Q_s$: simulated discharge; $Q_o$: observed discharge; | (Nash and Sutcliffe, 1970) |
| Ratio of the root mean square error and the standard deviation of observation (RSR) | $\dfrac{\sqrt{\sum_{t=1}^{N}\left(Q_{o,t} - Q_{s,t}\right)^2}}{\sqrt{\sum_{t=1}^{N}\left(Q_{o,t} - \bar{Q}_o\right)^2}}$ | $(0, +\infty)$ | 0 | $\bar{Q}_o$: mean of observed discharge; $\bar{Q}_s$: mean of simulated discharge; | (Moriasi et al., 2007) |
| Pearson's correlation coefficient ($r$) | $\dfrac{\sum_{t=1}^{N}\left(Q_{s,t} - \bar{Q}_s\right)\left(Q_{o,t} - \bar{Q}_o\right)}{\sqrt{\sum_{t=1}^{N}\left(Q_{s,t} - \bar{Q}_s\right)^2} - \sqrt{\sum_{t=1}^{N}\left(Q_{o,t}\right.}}$ | $(-1, 1)$ | 1 | $t$: sequence of the discharge series; | (Huang et al., 2012) |
| Modified Kling-Gupta efficiency (KGE) | $1 - \sqrt{(\alpha - 1)^2 + (\beta - 1)^2 + (r - 1}$ | $(-\infty, 1)$ | 1 | N: number of time steps; $\alpha$: ratio between the standard deviations of the simulated and observed data; $\beta$: ratio between the mean simulated and mean observed discharge | (King et al., 2012) |

**Table 5 Mean values of temperature, precipitation and simulated discharge in different scenarios**

| | | piControl scenario | Historical scenario | Future scenario | | | |
| --- | --- | --- | --- | --- | --- | --- | --- |
| | | | | RCP2.6 | RCP4.5 | RCP6.0 | RCP8.5 |
| Temperature (°C) | 1861-2005 | 6.40 | 6.53 | - | - | - | - |
| | 2006-2099 | 6.41 | - | 8.27 | 8.79 | 8.70 | 9.72 |
| | 2100-2299 | 6.43 | - | 8.38 | 10.48 | - | 19.94 |
| Precipitation (mm) | 1861-2005 | 821.8 | 805.7 | - | - | - | - |
| | 2006-2099 | 819.2 | - | 814.9 | 823.8 | 809.8 | 830.2 |
| | 2100-2299 | 835.7 | - | 854.2 | 841.0 | - | 790.4 |
| Discharge ($m^3s^{-1}$) | 1861-2005 | 10578.0 | 10294.4 | - | - | - | - |
| | 2006-2099 | 11338.6 | - | 10784.6 | 10592.6 | 10224.6 | 10617.8 |
| | 2100-2299 | 11698.5 | - | 11859.2 | 11824.3 | - | 10279.2 |

Table 4: Criteria of fit of the four hydrological models in the calibration period and in the wet and dry validation periods

Table 6 Performance of four hydrological models in the upper Yangtze River at the calibration period and the wet and dry validation periods

| Criterion | Threshold | Calibration/validation | HBV | SWAT | SWIM | VIC |
|---|---|---|---|---|---|---|
| NSE | >=0.7 | 1979-1990 | 0.86 | 0.81 | 0.75 | 0.89 |
| | | 1967-1978 (wet period) | 0.86 | 0.79 | 0.7 | 0.88 |
| | | 1991-2002 (dry period) | 0.86 | 0.81 | 0.75 | 0.89 |
| RSR | <=0.6 | 1979 - 1990 | 0.39 | 0.43 | 0.50 | 0.33 |
| | | 1967-1978 (wet period) | 0.38 | 0.46 | 0.55 | 0.34 |
| | | 1991-2002 (dry period) | 0.36 | 0.42 | 0.48 | 0.32 |
| $r$ | >=0.9 | 1979-1990 | 0.92 | 0.91 | 0.91 | 0.97 |
| | | 1967-1978 (wet period) | 0.92 | 0.90 | 0.89 | 0.96 |
| | | 1991-2002 (dry period) | 0.94 | 0.92 | 0.93 | 0.97 |
| KGE | >=0.7 | 1979-1990 | 0.87 | 0.9 | 0.7 | 0.71 |
| | | 1967-1978 (wet period) | 0.90 | 0.88 | 0.65 | 0.69 |
| | | 1991-2002 (dry period) | 0.85 | 0.89 | 0.56 | 0.68 |

Table 5: Comparison of changes in the mean annual discharge, Q10 and Q90 in the period 2006-2299 under the scenarios of anthropogenic climate change and the piControl scenario

| Dataset | Rate of change (Mean) (%) | Rate of change (Q10) (%) | Rate of change (Q90) (%) | Standard deviation | Coefficient of variation |
|---|---|---|---|---|---|
| piControl | - | - | - | 980.4 | 0.08 |
| RCP2.6 | -0.92 | -0.49 | -3.8 | 1146.2 | 0.10 |
| RCP4.5 | -4.7 | 5.2 | -11.9 | 1819.6 | 0.17 |
| RCP6.0 | -7.7 | -7.3 | -7.9 | 965.9 | 0.09 |
| RCP8.5 | -18.2 | -2.9 | -30.6 | 2347.9 | 0.25 |

**Table 7 Relative changes in mean annual discharge, Q10 and Q90 in the periods 2070 - 2099 and 2270 - 2299 under the scenarios of anthropogenic climate change relative to the piControl scenario**

| Period | Scenarios | Relative change of mean discharge (%) | Relative change of Q10 (%) | Relative change of Q90 (%) | Standard deviation | Coefficient of variation |
|---|---|---|---|---|---|---|
| | piControl | - | - | - | 607.1 | 0.05 |
| | RCP2.6 | -4.2 | -1.2 | -5.4 | 681.1 | 0.06 |
| 2070-2099 | RCP4.5 | -1.1 | 3.2 | -10.9 | 997.1 | 0.09 |
| | RCP6.0 | -9.1 | -3.5 | -10.6 | 763.7 | 0.07 |
| | RCP8.5 | -0.7 | 4.3 | -3.5 | 917.3 | 0.08 |
| | piControl | - | - | - | 767.6 | 0.06 |
| | RCP2.6 | 2.2 | 2.5 | 3.2 | 608.8 | 0.05 |
| 2270-2299 | RCP4.5 | 2.6 | 6.6 | -2.3 | 1255.9 | 0.11 |
| | RCP6.0 | - | - | - | - | - |
| | RCP8.5 | -30.6 | -13.2 | -50.4 | 1397.4 | 0.16 |

---

## Author Comment (AC2) · 22 Dec 2019

Dear reviewer

Thank you very much for providing comments. Your suggestions are helpful for improving the manuscript. In the following, we address all comments point-by-point.

> **Comment 1:** *The title could be edited more suitably for the main purpose of the paper. In the title, "hydrological modeling and GCMs" expressions could be used.*

**Answer:**

According to the opinions of reviewers, we have modified paper title to "A 439-year simulated daily discharge dataset (1861-2299) for the upper Yangtze River, China".

> **Comment 2:** *In the abstract; the objective, the original value and the widespread impact of the study could be expressed better, the introduction could be handled for meaning unity and logical continuity.*

**Answer:**

Thank for your suggestion, we tried to express better the objective, value and the widespraed impact of the study in the abstract and modified some sentences in the introduction.

1. Abstract:

"The long-term daily discharge dataset could be used in the international context and water management, e.g. in the framework of Inter-Sectoral Impact Model Intercomparison Project (ISIMIP) by providing clue to what extent human-induced climate change could impact streamflow and streamflow trend in the future."

2. We rewrote some sentences in the introduction:

"Global warming is the long-term rise in average temperature of the earth's climate system. Warming temperature alters global water circulation processes and could significantly influence the sustainability of society and economy (Jung et al., 2011). The variation in water resources availability in the context of global warming is acknowledged as a focus of many international research projects (Stagl et al., 2016; Raman et al., 2018; Maisa et al., 2019). The long-term accurate (as much as possible) daily discharge time series are crucial for in-depth understanding of the changes in streamflow, and they are needed for subsequent climate change impact studies. However, discharge is monitored usually only for short observational periods in most river basins.

For generation of the long-term streamflow series, many data mining techniques including the sedimentological method, the hydrological field survey method, and the documentary analysis method can be applied (Longfield et al., 2018). Nevertheless, low temporal resolution and insufficient accuracy of these estimations can hardly meet the demands of practical and research applications. Instead, the observed climatic variables and the outputs of climate models have often been used to drive hydrological models to evaluate changes in streamflow in the context of climate change (Braud et al., 2010; Chen et al., 2017; Su et al., 2017; Dahl, 2018; Seneviratne et al., 2018).

But there is lack of research on the quantitative estimation of long-term streamflow for period longer than 400 years under different scenarios with and without anthropogenic climate change (Meaurio, 2017).

The Yangtze River is the longest river in China. It originates from the Tibetan Plateau and enters the East China Sea after flowing through 11 provinces. With a large topographic gradient and substantial water supply of approximately 10,000 $m^3s^{-1}$ on the average, the upper Yangtze River is rich in hydropower resources, but subjected to destructive flash floods. The Yangtze River has the longest hydrological observations in China. Data provided by the Cuntan hydrological station, which started operating in 1939, facilitates hydro-meteorological studies in the instrumental period (Su et al., 2008; Wang et al., 2008; Su et al., 2017). As changes in streamflow at the Cuntan station directly influence inflow to the Three Gorges Reservoir, establishing long-term discharge series at the Cuntan station can support effective management of hydraulic projects. Besides, the longer discharge series can also provide a possibility to explore impacts of anthropogenic climate change on hydrology for international climate change research community. Therefore, we simulated daily discharge at the Cuntan hydrological station in the upper Yangtze River in the period 1861 - 2299 using available climate model outputs.

The outputs of four downscaled GCMs (GFDL-ESM2M, HadGEM2-ES, IPSL-CM5A-LR, and MIROC5) are utilized to drive four hydrological models (HBV, SWAT, SWIM and VIC) to simulate discharge at the Cuntan station. The climate forcing comprise (a) the scenario with anthropogenic climate change for the period 1861 - 2299, which is subdivided into the historical period (1861 - 2005) and the future period (2006 - 2299) under different Representative Concentration Pathways (RCPs), and (b) the preindustrial control scenario (piControl) for the period 1861 - 2299, which is used as a reference to detect the influence of anthropogenic climate change on streamflow in the upper Yangtze River."

***Comment 3:*** *In the section of study area, Fig. 1 must be pointed out and river basin characteristics (elevation, slope, aspect, land cover, soil type, etc.) must be given shortly to understand the hydrological cycle of the basin.*

**Answer:**

As suggested by the reviewer, the elevation, slope, aspect meteorological station, land cover and soil type are added to the Fig.1, and explanations are added to section of Study Area in the revised manuscript. The text is as follows:

"The catchment area of the Cuntan hydrological station (29 ° 37 ′ N, 106 ° 36 ′ E) in the upper Yangtze River is approximately 860,000 km$^2$, and 352.7 billion m$^3$ water is flowing through this point annually with average discharge of 109,34 m$^3$s$^{-1}$ in the period of instrumental measurements beginning in 1939. Location of the Cuntan hydrological station, 311 GCM grids, meteorological stations and spatial distribution of the land use and soil types in the upper Yangtze River basin are shown in Fig. 1. Prairie grassland and acid purple soil are the most widespread of land use and soil type in the upper Yangtze River basin. The upper Yangtze River has complex geomorphic types and broken topography. Mountains and plateaus can be found in most parts of the region, hills and plains are few. Influenced by the East Asia subtropical monsoon and a complex topography, climate varies across the basin with annual air temperature and precipitation being high in the southeast but low in the northwest headstream region. According to observational data, the areal averaged annual mean temperature and precipitation are 12.3 °C and 1018 mm, respectively, in the period 1961 - 2017 in the upper Yangtze River basin."

[Figure]

Figure 1 Location of the Cuntan hydrological station, GCM grids, meteorological stations and spatial distribution of the land use and soil types in the upper Yangtze River basin

***Comment 4:*** *In the methods, a flowchart included all processes could be given for the sake of clarity. Moreover, GIS and remote sensing techniques used could be mentioned shortly in this section.*

**Answer:**

We made a flowchart included all processes for the sake of clarity and added references about hydrological models. See in section 3.4 (page 4-5) as follow:

"Four hydrological models, HBV (Bergstrom et al., 1973), SWAT (Arnold et al., 1998), SWIM (Krysanova et al., 2005) and VIC (Liang et al., 1994) are used to simulate river discharge at the Cuntan hydrological station, and a flowchart of the hydrological modelling process is shown in Fig. 2."

[Figure]

Figure 2 Flowchart of the hydrological modelling process

We made an improper express and caused a misunderstanding about the sentence "The remote sensing data were assimilated to obtain monthly evapotranspiration with a spatial resolution of 0.25°". The remote sensing data is evapotranspiration data, which was produced by mean of remoting sensing techniques. In our study, we compare simulated outputs and GLEAM by means of GIS tools. See discription in section 3.3 as follow:

"Evapotranspiration data from the Global Land Evaporation Amsterdam Model (GLEAM) for 1986 - 2005 that were released by the University of Bristol (Miralles et al., 2011) are used in our study to cross-check the performances of the hydrological models by means of the geographic information system (GIS) tools. The GLEAM data was generated based on a variety of satellite-sensor products at monthly scale with a spatial resolution of 0.25°. The spatial distributions of simulated

evapotranspiration with that from GLEAM are compared by GIS techniques, and kappa value of confusion matrix is also applied to evaluate the accuracy of simulated evapotranspiration (taking VIC output as an example) by refer to GLEAM."

> ***Comment 5:*** *In section 3.1, some information about downscaling, bias correction and data assimilation processes (details, techniques, etc.) of GCMs should be given. And it should be explained that how GCMs grid data was inputted into the hydrological models (as lumped data?). In addition, more details about GCMs (model structures, etc.) and RCP scenarios should be presented.*

**Answer:**

1. The four GCMs used in this study were widely used in previous studies. The details of them can be obtained from the classical references: Taylor et al. (2012) for GFDL-ESM2M, Jones et al. (2011) for HadGEM2-ES, Dufresne et al. (2013) for IPSL-CM5A-LR and Watanabe et al. (2010) for MIROC5. As a result, we decided not to include a detailed description to keep the paper within a reasonable length.

2. The description of downscaling, bias correction and data assimilation processes can be found in section 3.1 Climate scenarios (page 3, lines 10-19). We also add references for them in case readers would be interested in detail. They are:

Frieler, K., Lange, S., Piontek, F., et al: Assessing the impacts of 1.5 °C global warming – simulation protocol of the Inter-Sectoral Impact Model Intercomparison Project (ISIMIP2b), Geosci. Model Dev., 10, 4321-4345, https://doi.org/10.5194/gmd-10-4321-2017, 2017.

Lange, S.: Bias correction of surface downwelling longwave and shortwave radiation for the EWEMBI dataset, Earth. Syst. Dynam., 9, 627-645, https://doi.org/10.5194/esd-2017-81, 2018.

3. GCMs data are inputted to the hydrological models using data in grid format based on the needs of hydrological models Since the main purpose of this paper is to discuss the discharge dataset and not hydrological modelling, we gave a brief introduction of 4 hydrological models (see Table 2). More details can be found in the Hattermann et al. (2017).

***Comment 6:*** *In section 3.2, please provide details (precipitation, temperature, etc.) of observed daily meteorological data and explain how the data of 189 ground-based stations was inputted into the hydrological models (mean or spatial interpolation/extrapolation of the data?). Furthermore, please give reference/references of flood events (1870 and 1931) because of measurements beginning in 1939.*

**Answer:**

1. The observed daily meteorological data are derived from the National Meteorological Information Centre of China Meteorological Administration (http://data.cma.cn/). The daily data from 189 ground-based stations are inputted into the hydrological models after interpolation. We added explanations in revised manuscript to make this clear.

"The observed daily meteorological data for 1951 - 2017 from 189 ground-based stations in the upper Yangtze River Basin used in this study were quality controlled by considering changes in instrument type, station relocations, and trace biases at the National Meteorological Information Centre of China Meteorological Administration (Ren et al., 2010), which was inputted into the hydrological models by spatial interpolation. During 1951 - 2017, annual precipitation shows a decreasing trend, with multi-year average of 935 mm, and annual mean temperature has shown a positive trend with multi-year average of 10.5 ℃."

2. Following references are added to cite the flood events (1870 and 1931) (Page 4, lines 5-10):

Changjiang Water Resources Commission of the Ministry of Water Resources: The flood and drought disasters in the Yangtze River Basin, China Water & Power Press, Beijing, China, 2002.

Hu, M.S. and Luo, C.Z: The historical flood of China, China Bookstore press, Beijing, China, 1992.

Luo, C.Z. and Le, J.X: The flood of China, China Bookstore press, Beijing, China, 1996.

***Comment 7:*** *In section 3.3, please give more details about GLEAM and the assimilation technique. And please present the comparison results of GLEAM and 4 hydrological models for internal consistency of the models in the results section.*

**Answer:**

1. Evapotranspiration data from the Global Land Evaporation Amsterdam Model (GLEAM) for 1986 - 2005 that were released by the University of Bristol (Miralles et al., 2011) is used to cross-check performances of the hydrological models. The GLEAM data was generated based on a variety of satellite-sensor products at monthly scale with a spatial resolution of 0.25°. For more details please refer to Miralles et al., 2011 (Section 3.3, page 4). We also added explanations in revised manuscript to clarify this.

2. Besides, we compare results of GLEAM and 4 hydrological models for checking the internal consistency of the models in the section 4.2 Calibration and validation of the hydrological models (Section 4.2, page 7 and Fig. 8) as follows:

"In addition, evapotranspiration outputs of HBV, SWAT, SWIM, VIC are compared with the GLEAM evapotranspiration data (see Section 3.3) in the period 1986 - 2005. The long-term averaged annual evapotranspiration simulated by the four models for the upper Yangtze River basin is 442 mm, 487 mm, 484 mm, 466 mm, respectively, quite close to the result from GLEAM (452 mm). The spatial patterns of the gridded evapotranspiration outputs of the HBV, SWAT, SWIM, VIC model and GLEAM all show low values in the northwest but high values in the southeast of the upper Yangtze River basin (Fig. 8). Furthermore, a matrix consisting of 500 randomly selected pixels from simulated evapotranspiration by VIC and corresponding GLEAM grids is set up to get the kappa value. The deduced kappa value of 0.62 indicates a substantial agreement of two date sources."

> ***Comment 8:*** *In section 3.4, please present input and output data of the models, model structures (loss and transform methods, soil moisture accounting, snow melting-accumulation, evapotranspiration, groundwater, canopy interception etc.) and parameter sets.*

**Answer:**

The input and output data of this study are the Global Climate Models (GFDL-ESM2M, HadGEM2-ES, IPSL-CM5A-LR and MIROC5) and simulated discharge (https://doi.org/10.4121/uuid:8658b22a-8f98-4043-9f8f-d77684d58cbc), respectively.

The hydrological processes including loss and transform methods, soil moisture accounting, snow melting and accumulation, groundwater-related processes, canopy interception etc, can be found in the references related with the HBV (Bergstrom et al., 1973), SWAT (Arnold et al., 1998), SWIM (Krysanova et al., 2005) and VIC (Liang et al., 1994). The references are included.

*Comment 9:* *In section 3.5, please indicate how many parameters were calibrated for each model and what is the range for each parameter.*

**Answer:**

Thank you for your suggestion. Ranges of parameters need to be calibrated are included in Table 3 to respond your request.

Table 3 The parameters and their ranges used for calibration of four hydrological models

| HBV | | SWAT | | SWIM | | VIC | |
|---|---|---|---|---|---|---|---|
| Name | Range | Name | Range | Name | Range | Name | Range |
| Threshold quick runoff (UZ1) | 0-100 | Deep aquifer percolation fraction ($Rchrg\_Dp$) | 0-1 | Routing coefficient 1 ($roc1$) | 1-100 | Non-linear baseflow begins ($Ds$) | 0-1 |
| Percolation to lower zone ($PREC$) | 0-6 | Saturated hydraulic conductivity ($Sol\_K$) | 0-100 | Routing coefficient 2 ($roc2$) | 1-100 | Maximum baseflow ($Ds_{max}$) | 0-30 |
| Non-linearity in soil water zone ($BETA$) | 1-5 | Maximum canopy storage ($Canmx$) | 0-10 | Evaporation coefficient ($thc$) | 0.5-1.5 | Maximum soil moisture ($Ws$) | 0-1 |
| Slow time constant upper zone ($KUZ1$) | 0.01-1 | Average slope steepness ($Slope$) | 0-0.6 | Baseflow factor for return flow travel time ($bff$) | 0.2-1 | Variable Infiltration Capacity curve ($b_i$) | 0-0.4 |
| Additional precipitation coefficient for snow at gauge ($SKORR$) | 1-3 | Available water capacity ($Sol\_Awc$) | 0-1 | Coefficient to correct channel width ($chwc0$) | 0.1-1 | Soil depth 1 ($d_1$) | 0.1-1 |
| Precipitation correction for rain ($PKORR$) | 0.8-3 | Initial SCS CN II value ($Cn2$) | 35-98 | Saturated conductivity ($sccor$) | 0.01-10 | Soil depth 2 ($d_2$) | 0.1-2 |
| | | Groundwater "revap" coefficient ($Gw\_Revap$) | 0.02-0.2 | Groundwater recession rate ($abf$) | 0.01-1 | Soil depth 3 ($d_3$) | 0.1-3 |
| | | Biological mixing efficiency ($Biomix$) | 0-1 | Initial conditions ($gwq0$) | 0.01-1 | | |
| | | Soil evaporation compensation factor ($Esco$) | 0-1 | Curve number ($cnum$) | 10-100 | | |

***Comment 10:*** *In section 3.6, please clarify the problem of "is the use of 1990 year land use convenient for particularly future scenarios?". In conclusions and discussion section, it is stated that 1990-year land use is one of the uncertainties and human interferences have escalated since the 1990s. In my opinion, using 1990 land use data is a big handicap of this study that is not convenient for future modeling. 1990 land use data can be considered suitable for only piControl and historical periods. Modeling of future flows by using the land cover of 1990, when the industrial development has not started, will have misleading results, especially in human impact scenarios. The use of 2019 year land cover data would be more appropriate when modeling the future and this will make the results more consistent. I don't accept the pretext of not available by reason of the fact that current land use data can be easily obtained from various sources*

**Answer:**

Thank you for your valuable comment and suggestions. We cannot deny that the newest land cover data may be more appropriate for simulating future discharge. However, it still cannot represent the land cover in the future. Therefore, we tried to use data, including land cover, reflecting the whole calibration/validation period, which would reduce uncertainty except for climate change. The calibration period is 1979-1990, thus, we used the 1990 land use as single geographical data to simulate the daily discharge.

In the previous study (Su et al., 2016, as follow) the 1990 land use map was used for simulating the discharge for the upper Yangtze River, China, and other previous studies showed that impact of land use on stream flow is rather small (Wang et al., 2017; Lu et al., 2015). These references are now included in the manuscript.

Su, B., Huang, J. L., Zeng, X. L., Gao, C., and Jiang, T.: Impacts of climate change on streamflow in the upper Yangtze River basin, Clim. Change, 141, 533-546, https://doi.org/10.1007/s10584-016-1852-5, 2017.

Wang H., Sun F. B., Xia J., Liu W. B.: Impact of LUCC on stream flow based on the SWAT model over the Wei River basin on the Loess Plateau in China, Hydrol. Earth Syst. Sci., 21, 1929-1945, 2017 www.hydrol-earth-syst-sci.net/21/1929/2017/ doi:10.5194/hess-21-1929-2017

Lu Z. X., Zou, S. B., Qin Z. D., Yang, Y. G., Xiao, H. L., Wei Y. P., Zhang K., and Xie J. L.: Hydrologic Responses to Land Use Change in the Loess Plateau: Case Study in the Upper Fenhe River Watershed, Advances in Meteorology. 2015. 1-10. 10.1155/2015/676030.

**Comment 11:** *In section 4.1 and 4.3, while determining the trends, was statistical trend analysis applied? Or how were the trends determined? By only visual technique?*

**Answer:**

We combine the statistical and visual techniques to explore the trends presented in sections 4.1 and 4.3. We added detail values (shown in table 5), which are described in sections 4.1 and 4.3. In order to visualize the precipitation, temperature and simulated discharge, we presented trends in figure format (Fig. 3, Fig. 4 and Fig. 9).

Table 5 Mean values of temperature, precipitation and simulated discharge in different scenarios

| | | piControl scenario | Historical scenario | Future scenario | | | |
|---|---|---|---|---|---|---|---|
| | | | | RCP2.6 | RCP4.5 | RCP6.0 | RCP8.5 |
| Temperature (℃) | 1861-2005 | 6.40 | 6.53 | - | - | - | - |
| | 2006-2099 | 6.41 | - | 8.27 | 8.79 | 8.70 | 9.72 |
| | 2100-2299 | 6.43 | - | 8.38 | 10.48 | - | 19.94 |
| Precipitation (mm) | 1861-2005 | 821.8 | 805.7 | - | - | - | - |
| | 2006-2099 | 819.2 | - | 814.9 | 823.8 | 809.8 | 830.2 |
| | 2100-2299 | 835.7 | - | 854.2 | 841.0 | - | 790.4 |
| Discharge ($m^3 s^{-1}$) | 1861-2005 | 10578.0 | 10294.4 | - | - | - | - |
| | 2006-2099 | 11338.6 | - | 10784.6 | 10592.6 | 10224.6 | 10617.8 |
| | 2100-2299 | 11698.5 | - | 11859.2 | 11824.3 | - | 10279.2 |

[Figure]

Figure 9 The annual mean discharge at the Cuntan station simulated by four hydrological models (HBV, SWAT, SWIM, and VIC) under the piControl scenario and scenarios with anthropogenic climate change effects (a-b)

N.B.: Because of taking up too many pages, we only include Fig. 9 here . See Fig. 3 and Fig. 4 in the revised paper.

***Comment 12:*** *In section 4.2, please explain what the model performance threshold is. Is it for good performance? Please give reference/references for it (Table 4). In addition, please comment why SWIM and VIC models had lower performance in the validation period. Moreover, please provide Q90 results in Fig. 5 which shows the scatter plots. So, in this figure, the coefficient of determination (R2) of 4 hydrological models should be given for significance of the scatter plots. Likewise, WLS values of extreme conditions should be presented for the calibration period. For this section, finally, please show the mean annual evapotranspiration outputs of the other three models based on sub-basins in Fig. 7 and compare all of the models with GLEAM based on statistical and/or mathematical criteria. It is better to see the all models in comparison. Even further, VIC and GLEAM could be compared based on grid by means of a confusion matrix.*

**Answer:**

We have added the thresholds of acceptance in Table 6 and the references in Table 4 as follow:

"In this study, four criteria, the NSE, RSR, Pearson's correlation coefficient (r) and KGE, are applied to the daily discharge series to evaluate the performance of hydrological models (Krysanova et al., 2018; Table 4). Thresholds of acceptance of four criteria are derived from the references (Nash and Sutcliffe, 1970; Moriasi et al., 2007; Huang et al., 2012; King et al., 2012)."

Table 6 Performance of four hydrological models in the upper Yangtze River at the calibration period and the wet and dry validation periods

| criterion | Thresholds of acceptance | calibration/validation | HBV | SWAT | SWIM | VIC |
|---|---|---|---|---|---|---|
| NSE | >=0.7 | 1979-1990 | 0.86 | 0.81 | 0.75 | 0.89 |
| | | 1967-1978 (wet period) | 0.86 | 0.79 | 0.7 | 0.88 |
| | | 1991-2002 (dry period) | 0.86 | 0.81 | 0.75 | 0.89 |
| RSR | <=0.6 | 1979 - 1990 | 0.39 | 0.43 | 0.50 | 0.33 |
| | | 1967-1978 (wet period) | 0.38 | 0.46 | 0.55 | 0.34 |
| | | 1991-2002 (dry period) | 0.36 | 0.42 | 0.48 | 0.32 |
| r | >=0.9 | 1979-1990 | 0.92 | 0.91 | 0.91 | 0.97 |
| | | 1967-1978 (wet period) | 0.92 | 0.90 | 0.89 | 0.96 |
| | | 1991-2002 (dry period) | 0.94 | 0.92 | 0.93 | 0.97 |
| KGE | >=0.7 | 1979-1990 | 0.87 | 0.9 | 0.7 | 0.71 |
| | | 1967-1978 (wet period) | 0.90 | 0.88 | 0.65 | 0.69 |
| | | 1991-2002 (dry period) | 0.85 | 0.89 | 0.56 | 0.68 |

Table 4 Evaluation criteria for testing simulation capacity of hydrological models

| Criterion | Formula | Range | Ideal value | Notation | Reference |
|-----------|---------|-------|-------------|----------|-----------|
| Nash-Sutcliffe efficiency (NSE) | $1 - \dfrac{\sum_{t=1}^{N} (Q_{s,t} - Q_{o,t})^2}{\sum_{t=1}^{N} (Q_{o,t} - \acute{Q}_o)^2}$ | $(-\infty, 1)$ | 1 | $Q_s$: simulated discharge; $Q_o$: observed discharge; | (Nash and Sutcliffe, 1970) |
| Ratio of the root mean square error and the standard deviation of observation (RSR) | $\dfrac{\sqrt{\sum_{t=1}^{N} (Q_{o,t} - Q_{s,t})^2}}{\sqrt{\sum_{t=1}^{N} (Q_{o,t} - \acute{Q}_o)^2}}$ | $(0, +\infty)$ | 0 | $\bar{Q}$; $\bar{Q}$ $\acute{Q}_o$: mean of observed discharge; | (Moriasi et al., 2007) |
| Pearson's correlation coefficient (r) | $\dfrac{\sum_{t=1}^{N}(Q_{s,t} - \acute{Q}_s)(Q_{o,t} - \acute{Q}_o)}{\sqrt{\sum_{t=1}^{N}(Q_{s,t} - \acute{Q}_s)^2} - \sqrt{\sum_{t=1}^{N}(Q_{o,t} - \acute{Q}_c}}$ | $(-1, 1)$ | 1 | $\acute{Q}_s$: mean of simulated discharge; $t$: sequence of the discharge series; | (Huang et al., 2012) |
| Modified Kling-Gupta efficiency (KGE) | $\dfrac{1}{-\sqrt{(\alpha - 1)^2 + (\beta - 1)^2 + (r - 1)^2}}$ | $(-\infty, 1)$ | 1 | N: number of time steps; $\alpha$: ratio between the standard deviations of the simulated and observed data; $\beta$: ratio between the mean simulated and mean observed discharge | (King et al., 2012) |

The parameters in hydrological models are changed during the calibration process within ranges indicated in Table 4 and iterated continuously. The parameters are not adjusted during the validation period. This is the reason why some models had lower performance in the validation period compared to the calibration period.

We have added the Q10 and Q90 scatter plots with the coefficient of determination ($R^2$) in Fig.6.

N.B.: The previous version of Fig. 6 included only Q10. Now both Q10 and Q90 are included.

[Figure]

Figure 6 Comparison of the simulated and observed Q10, Q90 percentiles at the Cuntan station in the calibration period 1979 - 1990 (a) and validation period 1967 - 1978 (b) and 1991 - 2002 (c)

The annual evapotranspiration outputs of HBV, SWAT, SWIM, VIC are compared with GLEAM (0.25°×0.25°). The visual techniques are used to compare spatial distribution pattern of evapotranspiration. Besides, output of VIC is selected to set up a confusion matrix to compare with GLEAM evapotranspiration in gridded scale. The confusion matrix is obtained using 500 randomly selected pixel samples of VIC and GLEAM datasets. The kappa value, which is the result of confusion matrix, reaches 0.62, and indicating a substantial agreement of two datasets (the process is shown in the table below).

N.B.: Previous version showed the resample result from 0.25°×0.25° to 0.5°×0.5°. Now the results with original resolution are presented in the revised version.

[Figure]

Figure 8 Spatial distribution of multi-year averaged annual evapotranspiration in the upper Yangtze River basin for 1986 - 2005: HBV output (a), SWAT output (b), VIC output (c), SWIM output (d) and GLEAM data(e)

Table of confusion matrix process of VIC and GLEAM (not shown in the revised paper)

| ClassValue | 159-247 | 248-340 | 341-426 | 427-490 | 491-540 | 541-581 | 582-651 | Total | U_Accuracy | Kappa |
|---|---|---|---|---|---|---|---|---|---|---|
| 171-283 | 50 | 19 | 0 | 0 | 0 | 0 | 0 | 69 | 0.72 | 0.00 |
| 284-373 | 2 | 32 | 8 | 0 | 0 | 0 | 0 | 42 | 0.76 | 0.00 |
| 374-442 | 0 | 3 | 55 | 15 | 2 | 4 | 0 | 79 | 0.70 | 0.00 |
| 443-496 | 0 | 0 | 10 | 54 | 10 | 4 | 2 | 80 | 0.68 | 0.00 |
| 497-559 | 0 | 0 | 1 | 12 | 37 | 16 | 0 | 66 | 0.56 | 0.00 |
| 560-607 | 0 | 0 | 0 | 4 | 14 | 56 | 16 | 90 | 0.62 | 0.00 |
| 608-664 | 0 | 0 | 0 | 3 | 7 | 11 | 52 | 74 | 0.70 | 0.00 |
| Total | 52 | 54 | 74 | 88 | 70 | 91 | 70 | 500 | 0.00 | 0.00 |
| P_Accuracy | 0.96 | 0.59 | 0.74 | 0.61 | 0.53 | 0.62 | 0.74 | 0.00 | 0.67 | 0.00 |
| Kappa | 0.00 | 0.00 | 0.00 | 0.00 | 0.00 | 0.00 | 0.00 | 0.00 | 0.00 | **0.62** |

***Comment 13:*** *In competing interests section, please give information of author contribution for each author in favor of justice.*

**Answer:**

As suggested by reviewer, we added the author contribution (Page9) in the manuscript as follows:
"Chao Gao, Buda Su, Qianyu Zha, Cai Chen and Gang Luo run the hydrological models. Chao Gao, Buda Su and Tong Jiang analysed results and draft the manuscript. Xiaofan Zeng, Jinlong Huang, Min Xiong and Liping Zhang assisted in the data processing. Valentina Krysanova provided guidance for the calibration/validation of the models and the description of results. All authors reviewed the resulting inventory and assisted with paper writing."

***Comment 14:*** *In Fig. 1, please present DEM to get to know the basin better.*
*And please specify how many grid cells are in the basin.*

**Answer:**

The same as for question 3, we added the DEM, and the 311 grids of climate models in the basin (Fig. 1).
Title of the Fig. 1 is changed to: Location of the Cuntan hydrological station, GCM grids, meteorological stations and spatial distribution of the land use and soil types in the upper Yangtze River basin"

***Comment 15:*** *In titles of Fig. 2 and Fig. 3, "seasonal" expressions should be replaced with "monthly".*

**Answer:**

We have switched the "seasonal" to "monthly" in Fig. 3 and Fig. 4:

"Figure 3 Inter-annual (a) and long-term averaged monthly dynamics (b-d) of the surface air temperature in the upper Yangtze River basin: comparison of the piControl scenario with the anthropogenic climate change scenarios (periods: a: 1861 - 2299; b: 1861 - 2005; c: 2006 - 2099; and d: 2100 - 2299)

Figure 4 Inter-annual (a-b) and long-term averaged monthly dynamics (c-e) of precipitation in the upper Yangtze River basin: comparison of the piControl scenario with the anthropogenic climate change scenarios (periods: a: 1861-2299; b: 1861-2299; c: 1861 - 2005; d: 2006 - 2099; and e: 2100 - 2299)"

***Comment 16:*** *Although the study is daily flow modeling, the graphical representation in Fig. 6 is given on monthly time scale since the application period is very long. If so, please express this.*

**Answer:**

We selected monthly time scale to show trend because the period is quite long.

***Comment 17:*** *In Fig. 8, are the values mean outputs of hydrological models and GCMs (a-b)? And there is a noteworthy decreasing in discharge (d), why? Please comment. And why is "2100-2169" period not available (c-d)?*

**Answer:**

Fig.8 shows the simulated annual mean discharge averaged over four hydrological models and four GCMs.

The decrease of simulated discharge in 2170-2199 can be explained by trends of precipitation projected by GCMs, which show an increase trend in the 21[st] century but a decrease trend in the 22[nd] century.

There is no sense to compare only the long-term monthly means and average values for such a long period. Climate projections and hydrological impacts for the past and future periods can be analyzed for 30 y periods. Thus, we selected the periods of 2070-2099 and 2270-2299 to describe the average monthly discharge and statistical distribution of the daily maximum discharge:

"Similar to precipitation and temperature, average monthly discharge in 2070 - 2099 and 2270 - 2299 under both the piControl and RCP scenarios show single peak. Under RCP 4.5, a higher flood volume of August is projected in periods of 2070 - 2099 and 2270 - 2299 than the piControl scenario. Meanwhile, a higher volume in 2070 - 2099 but a lower in 2270 - 2299 under RCP8.5 is projected. Under RCP2.6, the flood volume of August is similar to piControl in both periods (Fig. 10a-b). The Generalized Logistic Distribution (GLD), which is the optimistic distribution by Kolmogorov - Smirnov goodness of fit test, is applied to describe the statistical distribution of the daily maximum discharge (represented by annual Q10) for 2070-2099 and 2270-2299. It is found that the return

level of daily maximum discharge under RCP2.6, RCP4.5, RCP6.0 and RCP8.5 are higher than piControl scenario in 2070 - 2099 (Fig. 10c). Under RCP 4.5, a higher average of return level of daily maximum discharge is projected in periods of 2070 - 2099 and 2270 - 2299 than the piControl scenario. For RCP8.5, the average of return level of daily maximum discharge is higher in 2070 - 2099 but lower in 2270 - 2299 than piControl scenario. Under RCP2.6, the average of return level of daily maximum discharge is similar to piControl scenario in both periods (Fig. 10c-d)"

> **Comment 18:** *In Table 4, it is interesting that the performance criteria are so close between the cal-val periods. Please comment what the reason.*

**Answer:**

Usually, a hydrological model has a certain stability when applied in the same watershed, and the criteria values in the validation period are lower than those in the calibration period but could be also close to them.

**Technical Corrections**

*Comment 19: P3L6 and L7: "0.2 °C/10a" and "-11 mm/10a" → Please present as open.*

**Answer:**

Thank you for your advice. We have adjusted it.

*Comment 20: P4L27: "(Moriasi et al., 2007)" must be placed in the end of the sentence.*

**Answer:**

We have moved the reference (Moriasi et al., 2007) to the end of sentence as follow:
"For evaluating daily hydrograph simulation, ratio of the root mean square error to the standard deviation of measured data (RSR) is recommended (Moriasi et al., 2007)"

*Comment 21: P5 L21-L22 and P6 L2-L3: "seasonal" → "monthly".*

**Answer:**

Thank you for your advice. Revised as suggested.

*Comment 22: P6 L27: "models could reproduce the monthly river flow …" → Please explain which one of "monthly modeling" or "daily modeling and transforming into monthly" was performed.*

**Answer:**

All four hydrological models simulated daily river flow, which was later transformed in the monthly river flow.

**Comment 23:** *P7 L13 and L14: "with the exception of RCP 6.0" → Please comment.*

**Answer:**

No projection has been done for 2170-2199 under RCP6.0 (see details in Table 1)

Table 1 Availability of climate scenarios from four GCMs for different periods

| Climate scenario | $CO_2$ concentration | GFDL-ESM2M | HadGEM2-ES | IPSL-CM5A-LR | MIROC5 |
|---|---|---|---|---|---|
| piControl scenario | 286 ppm | 1861-2099 | 1861-2299 | 1861-2299 | 1861-2299 |
| Historical scenario | Recorded $CO_2$ | 1861-2005 | 1861-2005 | 1861-2005 | 1861-2005 |
| | RCP2.6 | 2006-2099 | 2006-2299 | 2006-2299 | 2006-2299 |
| Future scenario | RCP4.5 | 2006-2099 | 2006-2099 | 2006-2299 | 2006-2099 |
| | RCP6.0 | 2006-2099 | 2006-2099 | 2006-2099 | 2006-2099 |
| | RCP8.5 | 2006-2099 | 2006-2099 | 2006-2299 | 2006-2099 |

**Comment 24:** *P7 L14 and L15: "an increasing trend in the 21st century, turning into a decreasing trend in the 22nd century." → Why? Please explain considering the precipitation.*

**Answer:**

It related to the precipitation trend, which increases in the 21st century but decreases in the 22nd centur*y*, as explained before (response to Q16). More details are added in the revision to make it clear:

"The simulated discharge time series for 1861 - 2299 under the piControl scenario without anthropogenic climate change and scenarios with anthropogenic climate change effects are shown in Fig. 9a-b. Similar to precipitation trend, annual mean discharge at the Cuntan station shows no significant trend from 1861 to 2299 under the piControl scenario. In historical period, annual mean discharge has shown a slight decrease trend in 1861 - 2005. Under RCPs, annual mean discharge will be in a significant upward trend by the end of the 21$^{st}$ century with increasing variation in the upper Yangtze River. Annual mean discharge shows no significant change since 2100 under RCP2.6 and RCP4.5, but a rapid decline is projected under high emission RCP8.5 scenario in future (Fig. 9a-b, Table 5)"

***Comment 25:*** *P7 L18: "11,517 m3s-1" → "11,517 m3s-1 (363.2 billion m3)".*
*For comparison of observation value stated in the section of study area.*

**Answer:**

According to the simulation results outputted by four hydrological models, the mean annual discharge in the piControl scenario is 11,517 $m^3s^{-1}$(the number represent the scenario of PiControl spans more than 400 years since 1861 to 2299). The value mentioned in the study is the mean value of the observations in recent decades, which is far less than the piControl. It is no sense to compare the piControl result and observation data. We revised the manuscript as:

"Comparison of relative changes in mean annual discharge for 2070-2099 and 2270-2299 under RCPs with that of the piControl scenario is presented in Table 7. Relative to the piControl scenario, change of annual mean discharge will be -4.2 %, -1.1 %, -9.1 % and -0.7 % respectively, under RCP2.6, RCP4.5, RCP6.0 and RCP8.5, in 2070 - 2099. And the relative change of annual mean discharge will be 2.2 %, 2.6 % and -30.6 %, respectively, under RCP2.6, RCP4.5 and RCP8.5 in 2270 - 2299 (Table 7)."

Table 7 Relative changes in mean annual discharge, Q10 and Q90 in the periods 2070 - 2099 and 2270 - 2299 under the scenarios of anthropogenic climate change relative to the piControl scenario

| Period | Scenarios | Relative change of mean discharge (%) | Relative change of Q10 (%) | Relative change of Q90 (%) | Standard deviation | Coefficient of variation |
|---|---|---|---|---|---|---|
| 2070-2099 | piControl | - | - | - | 607.1 | 0.05 |
| | RCP2.6 | -4.2 | -1.2 | -5.4 | 681.1 | 0.06 |
| | RCP4.5 | -1.1 | 3.2 | -10.9 | 997.1 | 0.09 |
| | RCP6.0 | -9.1 | -3.5 | -10.6 | 763.7 | 0.07 |
| | RCP8.5 | -0.7 | 4.3 | -3.5 | 917.3 | 0.08 |
| 2270-2299 | piControl | - | - | - | 767.6 | 0.06 |
| | RCP2.6 | 2.2 | 2.5 | 3.2 | 608.8 | 0.05 |
| | RCP4.5 | 2.6 | 6.6 | -2.3 | 1255.9 | 0.11 |
| | RCP6.0 | - | - | - | - | - |
| | RCP8.5 | -30.6 | -13.2 | -50.4 | 1397.4 | 0.16 |

***Comment 26:*** *P7 L23: "(except for RCP 4.5)" → Why? Please interpret.*

**Answer:**

Further evaluation needs to be done for some more specific reasons. In our opinion, it may be related to the Availability of time series (see details in Table 1). It contains data for 2006-2099 both for the RCPs and piControl. But there are different time series data of GCMs during the period of the 2100-2299, for example, for RCP2.6 (3 GCMs), RCP4.5 (only one, IPSL), RCP6.0 (none), RCP8.5 (2 GCMs). And now we divided 2006-2299 into 2006-2099 and 2100-2299 (page 7, line 4) to describe the relative changes in mean annual discharge for 2006-2099 and 2100-2299 under RCPs with that of the piControl scenario.

Table 1 Availability of climate scenarios from four GCMs for different periods

| Climate scenario | $CO_2$ concentration | GFDL-ESM2M | HadGEM2-ES | IPSL-CM5A-LR | MIROC5 |
|---|---|---|---|---|---|
| piControl | 286 ppm | 1861-2099 | 1861-2299 | 1861-2299 | 1861-2299 |
| Historical | Recorded $CO_2$ | 1861-2005 | 1861-2005 | 1861-2005 | 1861-2005 |
| Future | RCP2.6 | 2006-2099 | 2006-2299 | 2006-2299 | 2006-2299 |
| | RCP4.5 | 2006-2099 | 2006-2099 | 2006-2299 | 2006-2099 |
| | RCP6.0 | 2006-2099 | 2006-2099 | 2006-2099 | 2006-2099 |
| | RCP8.5 | 2006-2099 | 2006-2099 | 2006-2299 | 2006-2099 |

***Comment 27:*** *P8 L30: "For the calibration" → "For the calibration and the validation periods".*

**Answer:**

We corrected the statement and deleted the sentence "For the calibration and the validation periods".

"Four criteria, including the NSE, KGE, RSR and r, are used to evaluate the parameterization results. To assess the models' ability to satisfactorily simulate discharge under different climate conditions, hydrological models are validated both in dry and wet periods. Besides, evapotranspiration outputs by simulation process are compared with remote-sensing-based evapotranspiration from the GLEAM dataset to further validate performance of the models."

*Comment 28: P9 L1: "a cross-validation method" → Please give info. Leave-one-out or k-fold technique? If leave-one-out technique, it is problem of independence of validation dataset.*

**Answer:**

Here, the 'cross-validation' is not another technique but a method that considers evapotranspiration apart from discharge to test performance of hydrological model and improve the reliability of results. "Four criteria, including the NSE, KGE, RSR and r, are used to evaluate the parameterization results. To assess the models' ability to satisfactorily simulate discharge under different climate conditions, hydrological models are validated both in dry and wet periods. Besides, evapotranspiration outputs by simulation process are compared with remote-sensing-based evapotranspiration from the GLEAM dataset to further validate performance of the models."

*Comment 29: P9 L6 and L7: "the simulated extreme peak values in the 1930s, 1950s and 1990s were also in good agreement with the historical documented records ...→Please add "except 1998 flood" for objectiveness and being non-manipulative. It is specified that the modeled peak flow is 36,000 m3/s (in P6 L20) and the observed value is 68,500 m3/s (in P4 L9) for the 1998 flood event.*

**Answer:**

Thank you for your vital advice. We mentioned in the paper that the peak flows of simulated discharge in 1930s, 1950s and 1990s were 64,300 $m^3s^{-1}$, 53,900 $m^3s^{-1}$ and 60,700 $m^3s^{-1}$, respectively, deviating by less than 10 % from the recorded peaks, i.e., it means we compared the daily peak flows simulated by hydrological models and recorded peaks, rather than the Q10 results. Now, we reorganized the statement of this paper as follows:

"The four hydrological models can also properly simulate high flow and low flow represented by Q10 and Q90 in calibration and validation periods. For example, Q10 result illustrates that the several severe floods mentioned previously are reproduced quite well by the model simulations: the peak flows of simulated discharge were 64,300 $m^3s^{-1}$, 53,900 $m^3s^{-1}$ and 60,700 $m^3s^{-1}$, respectively, in the 1930s, 1950s and 1990s, deviating by less than 10 % from the recorded peaks (Fig. 6)."

*Comment 30: The references in P12 L5 and P12 L8 must be replaced due to the alphabetical order.*

**Answer:**

Thanks for your suggestion. Revised as suggested.

***Comment 31:*** *In accordance with the aim-scope of the journal, the dataset control process of the review is rather important case. For future reuse and reinterpretation, I checked the quality of the datasets which are available at the relevant web link. I think there is potential of the data being useful in the future but there is only simulated discharge file (xls) as time series. The observed data (discharge, precipitation, temperature, etc.), the data of GCMs (precipitation, temperature, etc. for pi and RCP scenarios), the evapotranspiration data of 4 hydrological models and GLEAM (as time series), the grid evapotranspiration data (GLEAM and VIC) and the other grid data used (DEM, soil and land use data) must be also presented for both reproducibility of scientific and usefulness of data. In addition, to perform tests for data quality, the above-mentioned data sets are necessary. The availability of these datasets is important for usefulness and completeness, too. And at the aim and scope web page of the journal, the expression of "each article should publish as much data as possible" supports the completion of deficiencies.*

**Answer:**

We will upload the simulated discharge file and evapotranspiration data, observed meteorological data including precipitation and temperature, etc. GCMs data, GLEAM evapotranspiration, 1990 land use, DEM at 90m resolution. Amongst, GCM data (precipitation, temperature, etc. for pi and RCP scenarios) were obtained from ISMIP project. 
[revised manuscript text omitted]
}\left(Q_{s,t} - Q_{o,t}\right)^2}{\sum_{t=1}^{N}\left(Q_{o,t} - \bar{Q}_o\right)^2}$ | $(-\infty, 1)$ | 1 | $Q_s$: simulated discharge; $Q_o$: observed discharge; $\bar{Q}_o$: mean of observed discharge; |
| Ratio of the root mean square error and the standard deviation of observation (RSR) | $\dfrac{\sqrt{\sum_{t=1}^{N}\left(Q_{o,t} - Q_{s,t}\right)^2}}{\sqrt{\sum_{t=1}^{N}\left(Q_{o,t} - \bar{Q}_o\right)^2}}$ | $(0, +\infty)$ | 0 | $\bar{Q}_s$: mean of simulated discharge; |
| Pearson's correlation coefficient ($r$) | $\dfrac{\sum_{t=1}^{N}\left(Q_{s,t} - \bar{Q}_s\right)\left(Q_{o,t} - \bar{Q}_o\right)}{\sqrt{\sum_{t=1}^{N}\left(Q_{s,t} - \bar{Q}_s\right)^2} - \sqrt{\sum_{t=1}^{N}\left(Q_{o,t} - \bar{Q}_o\right)^2}}$ | $(-1, 1)$ | 1 | $t$: sequence of the discharge series; N: number of time steps; |
| Modified Kling-Gupta efficiency (KGE) | $1 - \sqrt{(\alpha - 1)^2 + (\beta - 1)^2 + (r - 1)^2}$ | $(-\infty, 1)$ | 1 | $\alpha$: ratio between the standard deviations of the simulated and observed data; $\beta$: ratio between the mean simulated and mean observed discharge |

[revised manuscript text omitted]